# Long-term risk of psychiatric disorder and psychotropic prescription after SARS-CoV-2 infection among UK general population

Yunhe Wang [1,10], Binbin Su [2,10], Junqing Xie [3,4,5] ✉, Clemente Garcia-Rizo [6,7,8] & Daniel Prieto-Alhambra [3,9]

Despite evidence indicating increased risk of psychiatric issues among COVID-19 survivors, questions persist about long-term mental health outcomes and the protective effect of vaccination. Using UK Biobank data, three cohorts were constructed: SARS-CoV-2 infection ($n = 26{,}101$), contemporary control with no evidence of infection ($n = 380{,}337$) and historical control predating the pandemic ($n = 390{,}621$). Compared with contemporary controls, infected participants had higher subsequent risks of incident mental health at 1 year (hazard ratio (HR): 1.54, 95% CI 1.42–1.67; $P = 1.70 \times 10^{-24}$; difference in incidence rate: 27.36, 95% CI 21.16–34.10 per 1,000 person-years), including psychotic, mood, anxiety, alcohol use and sleep disorders, and prescriptions for antipsychotics, antidepressants, benzodiazepines, mood stabilizers and opioids. Risks were higher for hospitalized individuals (2.17, 1.70–2.78; $P = 5.80 \times 10^{-10}$) than those not hospitalized (1.41, 1.30–1.53; $P = 1.46 \times 10^{-16}$), and were reduced in fully vaccinated people (0.97, 0.80–1.19; $P = 0.799$) compared with non-vaccinated or partially vaccinated individuals (1.64, 1.49–1.79; $P = 4.95 \times 10^{-26}$). Breakthrough infections showed similar risk of psychiatric diagnosis (0.91, 0.78–1.07; $P = 0.278$) but increased prescription risk (1.42, 1.00–2.02; $P = 0.053$) compared with uninfected controls. Early identification and treatment of psychiatric disorders in COVID-19 survivors, especially those severely affected or unvaccinated, should be a priority in the management of long COVID. With the accumulation of breakthrough infections in the post-pandemic era, the findings highlight the need for continued optimization of strategies to foster resilience and prevent escalation of subclinical mental health symptoms to severe disorders.

The continuing spread of coronavirus disease 2019 (COVID-19) caused by severe acute respiratory syndrome coronavirus 2 (SARS-CoV-2) remains a major public health concern and results in enormous disease burden, with more than 774.7 million cases and 7.0 million deaths registered worldwide as of 18 February 2024 (ref. 1). Emerging evidence exists for the direct (through infection) and indirect (through change in environmental stressors and individual behaviours) effects of SARS-CoV-2 on the pulmonary and multiple extrapulmonary organs, including the metabolic, renal and cardiovascular systems, during and beyond the acute phase of COVID-19 of any severity[2–4].

**Fig. 1 | Flowchart of study design and cohort construction.**

Studies have also reported an increased risk of neurological and psychiatric disorders in individuals admitted to hospital for COVID-19 and those with mild or asymptomatic disease during 3 to 12 months after infection[5–9]. However, these studies have so far been based on electronic health record (EHR) or registry data that are less representative of the general population (recruitment is dependent on health system utilization) and more vulnerable to latent bias, confounding and the potential impact of the disruptions in healthcare services during the pandemic compared with prospective cohort based on the general population. Notably, although EHR-based studies used different settings of negative control groups including individuals with influenza and other diseases, socioeconomic and lifestyle factors and public health interventions in the context of the pandemic (such as vaccination) that were associated with both SARS-CoV-2 infection and mental health conditions were only crudely measured or not available in these analyses[5–8], leading to latent residual confounding and making causal interpretation of the findings challenging. Although studies suggested that vaccination before infection only partly mitigates the risk of psychiatric sequelae during a maximum of 6 months' follow-up[10,11], the effect of vaccination over longer follow-up periods, such as 1 year, remains unclear. Random, well-controlled, cohort studies based on the general population with detailed and robust recording of confounding factors and long-term follow-up may be less subject to potential bias and confounding than registry data, and are urgently needed to improve the current understanding of the long-term psychiatric sequelae of COVID-19 and the potential protective effect of vaccination.

In this study, we use prospective data from UK Biobank to quantify the incidence and relative risk of psychiatric diagnoses and related psychotropic prescriptions in participants who had a positive test for SARS-CoV-2 during a 1-year follow-up period after SARS-CoV-2 infection.

We explored whether the association between COVID-19 and the subsequent psychiatric outcomes observed in previous EHR-based studies varied by test setting of SARS-CoV-2 infection and vaccination status.

## Results

The study design and the process of cohort construction are shown in Fig. 1. The primary cohorts comprised 26,101 participants in the SARS-CoV-2 infection group, 380,337 in the contemporary control group and 390,621 in the historical control group. Median follow-up for any psychiatric diagnosis, any psychotropic prescription and any mental health outcome was 365, 257 and 256 days, respectively. The demographic and medical characteristics of the infection, contemporary control and historical control groups after weighting are shown in Table 1, and the characteristics of three groups before weighting are shown in Supplementary Table 2. Characteristics of participants without a history of any mental health outcome in the past 2 years before the start of follow-up before and after weighting are shown in Supplementary Tables 3 and 4, respectively. The number of individuals with a history of mental health outcome in the past 2 years before the start of follow-up who were excluded from the incident analyses between the infection group and the contemporary control is provided in Supplementary Table 5.

### Risk of mental health outcomes after SARS-CoV-2 infection
**COVID-19 group versus contemporary control.** Before weighting, participants in the infection group were younger (mean age: 66.0 vs 68.8 years), less likely from the White ethnic group (84.6 vs 93.7%), more socioeconomically deprived (mean index of multiple deprivation (IMD): 20.5 vs 17.3) and more physically obese (mean body mass index (BMI): 28.1 vs 27.3) than the contemporary control group

**Table 1 | Demographic and medical characteristics of SARS-CoV-2 infection, contemporary control and historical control groups after weighting**

| Characteristics | SARS-CoV-2 infection (n=26,101) | Contemporary control (n=380,337) | Historical control (n=390,621) | ASMD between infection and contemporary control* | ASMD between infection and historical control* |
|---|---|---|---|---|---|
| Age, mean (s.d.) | 68.5 (8.4) | 68.6 (8.1) | 66.9 (8.2) | 0.02 | 0.06 |
| Sex, male (%) | 11,824 (45.3) | 170,011 (44.7) | 175,779 (45.0) | 0.01 | 0.02 |
| Ethnicity, White (%) | 24,117 (92.4) | 354,094 (93.1) | 364,059 (93.2) | 0.03 | 0.03 |
| Index of multiple deprivation, mean (s.d.) | 18.2 (13.7) | 17.5 (13.9) | 17.6 (13.9) | 0.05 | 0.07 |
| Body mass index, mean (s.d.) | 27.5 (4.6) | 27.4 (4.7) | 27.4 (4.7) | 0.02 | 0.03 |
| Current smoker (%) | 2,688 (10.3) | 36,893 (9.7) | 38,671 (9.9) | 0.02 | 0.03 |
| Current drinker (%) | 23,830 (91.3) | 349,530 (91.9) | 358,590 (91.8) | 0.02 | 0.03 |
| Physical activity, high level (%)# | 8,457 (32.4) | 124,751 (32.8) | 127,733 (32.7) | 0.01 | 0.02 |
| Vaccination status, fully vaccinated (%) | 10,832 (41.5) | 148,331 (39.0) | NA | 0.04 | NA |
| Medications (%)† | | | | | |
| Lipid lowering drugs | 9,527 (36.5) | 135,020 (35.5) | 133,202 (34.1) | 0.02 | 0.03 |
| RAS inhibitors | 6,395 (24.5) | 90,901 (23.9) | 91,015 (23.3) | 0.02 | 0.02 |
| Other anti-hypertensives | 2,949 (11.3) | 41,076 (10.8) | 41,796 (10.7) | 0.02 | 0.02 |
| Anticoagulants | 1,175 (4.5) | 16,354 (4.3) | 15,234 (3.9) | 0.01 | 0.04 |
| Antiplatelet drugs | 3,080 (11.8) | 42,978 (11.3) | 42,187 (10.8) | 0.01 | 0.03 |
| Proton pump inhibitors | 7,935 (30.4) | 111,819 (29.4) | 112,889 (28.9) | 0.02 | 0.04 |
| Diabetes medicines | 1,905 (7.3) | 25,863 (6.8) | 26,172 (6.7) | 0.02 | 0.03 |
| Systemic glucocorticoids | 1,436 (5.5) | 20,158 (5.3) | 24,609 (6.3) | 0.01 | 0.02 |
| Immunosuppressants | 339 (1.3) | 4,564 (1.2) | 4,687 (1.2) | 0.01 | 0.01 |
| Antineoplastic agents | 26 (0.1) | 380 (0.1) | 391 (0.1) | 0.01 | 0.01 |
| Coexisting conditions (%)† | | | | | |
| Acquired immunodeficiency syndrome | 26 (0.1) | 380 (0.1) | 391 (0.1) | 0.01 | 0.01 |
| Cancer | 2,949 (11.3) | 41,837 (11.0) | 41,796 (10.7) | 0.01 | 0.01 |
| Cerebrovascular disease | 626 (2.4) | 8,748 (2.3) | 8,203 (2.1) | 0.01 | 0.01 |
| Chronic obstructive pulmonary disease | 4,515 (17.3) | 63,516 (16.7) | 63,281 (16.2) | 0.02 | 0.02 |
| Chronic kidney disease | 1,488 (5.7) | 20,158 (5.3) | 19,531 (5.0) | 0.02 | 0.01 |
| Congestive heart failure | 496 (1.9) | 6,846 (1.8) | 6,641 (1.7) | 0.01 | 0.01 |
| Dementia | 235 (0.9) | 3,423 (0.9) | 2,734 (0.7) | 0.01 | 0.02 |
| Diabetes (uncomplicated) | 2,688 (10.3) | 36,512 (9.6) | 39,842 (10.2) | 0.02 | 0.02 |
| Diabetes (end-organ damage) | 835 (3.2) | 11,790 (3.1) | 11,719 (3.0) | 0.01 | 0.01 |
| Hemiplegia | 26 (0.1) | 380 (0.1) | 391 (0.1) | 0.01 | 0.01 |
| Liver disease | 209 (0.8) | 2,663 (0.7) | 2,734 (0.7) | 0.01 | 0.01 |
| Peptic ulcer | 653 (2.5) | 9,128 (2.4) | 8,984 (2.3) | 0.01 | 0.01 |
| Rheumatoid arthritis | 757 (2.9) | 11,410 (3.0) | 10,547 (2.7) | 0.01 | 0.01 |
| Blood pressure, mean (s.d.), mm Hg | | | | | |
| Systolic blood pressure | 139.2 (19.4) | 139.4 (19.2) | 139.5 (19.2) | 0.01 | 0.01 |
| Diastolic blood pressure | 82.1 (10.5) | 82.1 (10.5) | 82.1 (10.5) | 0.01 | 0.01 |
| Hospital admissions, mean (s.d.)† | 0.43 (1.75) | 0.38 (2.23) | 0.44 (2.22) | 0.02 | 0.01 |

#Physical activity status was measured using the International Physical Activity Questionnaire (IPAQ). †Data collected within past 1 year of $T_0$ from primary care records. *ASMD ≤ 0.10 is considered good balance between comparison groups.

(Supplementary Table 2). After weighting, all characteristics were well balanced between the two comparison groups (absolute standardized mean difference (ASMD) < 0.1) and index dates were fully aligned (Table 1 and Supplementary Fig. 2). The incidence and risk of the incident psychiatric diagnoses and psychotropic prescriptions in these groups are provided in Fig. 2. The incidence and risk of incident or recurrent mental health outcomes are shown in Supplementary

Fig. 3. The incidence and risk of composite mental health outcomes are provided in Fig. 3.

*Psychotic, mood or anxiety disorders.* Compared with the contemporary control group, participants in the infection group were at an increased risk of incident psychotic disorders (hazard ratio (HR): 1.83, 95% confidence interval (CI): 1.11–3.00, *P* = 0.017; difference in

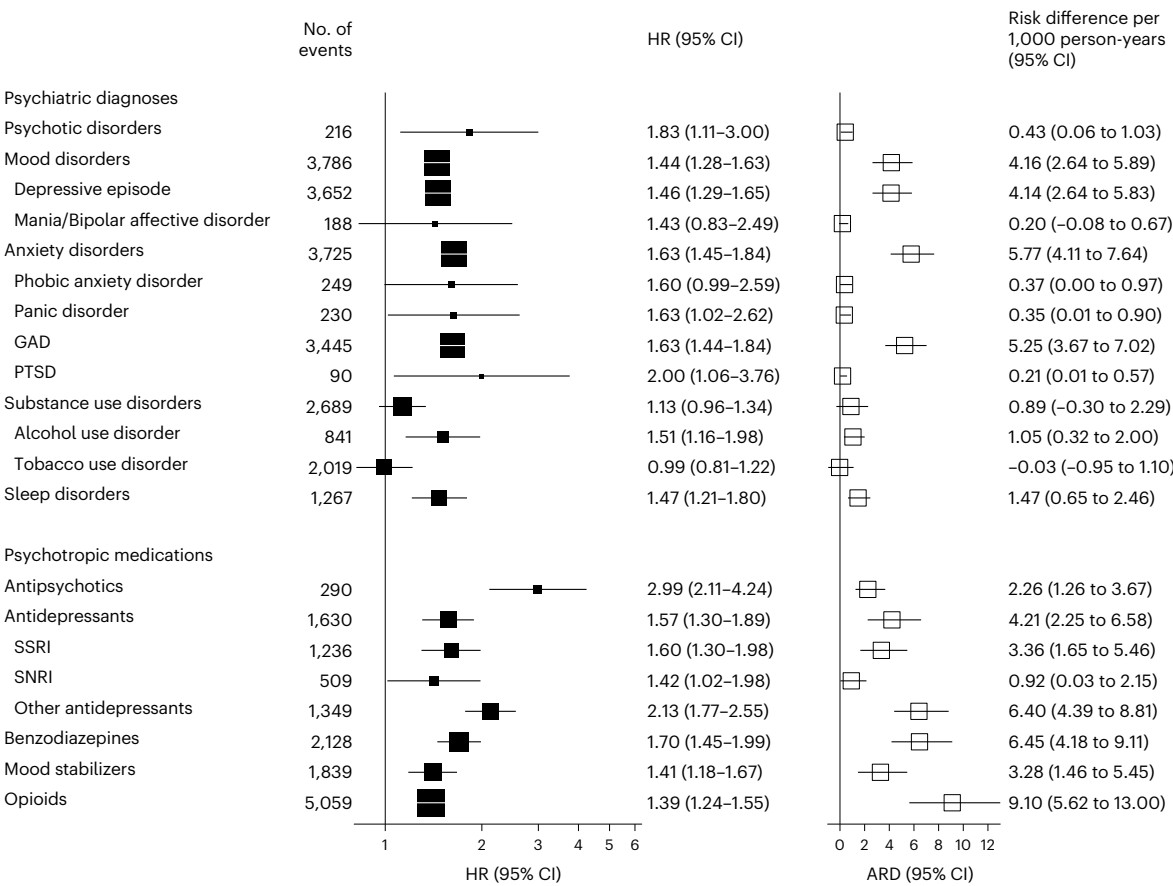

**Fig. 2 | Risks of incident psychiatric diagnoses and psychotropic prescriptions after SARS-CoV-2 infection compared with the contemporary control group.** Mental health outcomes were ascertained after SARS-CoV-2 infection until the end of follow-up. Results compare the SARS-CoV-2 infection group ($n$ = 19,353) to the contemporary control group with no evidence of infection ($n$ = 301,398). HRs were adjusted for predefined and data-driven covariates. Risk was reported in relative scale (HRs and 95% CIs) and absolute scale (absolute risk difference (ARD) per 1,000 person-years with 95 CIs). Squares represent estimates of HRs (with area inversely proportional to the variance) or risk difference, and error bars represent the corresponding 95% CIs. GAD, generalized anxiety disorder; PTSD, post-traumatic stress disorder.

incidence rate: 0.43, 95% CI: 0.06–1.03 per 1,000 person-years), mood disorders (1.44, 1.28–1.63, $P$ = 1.78 × 10$^{-9}$; 4.16, 2.64–5.89 per 1,000 person-years) and anxiety disorders (1.63, 1.45–1.84, $P$ = 4.41 × 10$^{-16}$; 5.77, 4.11–7.64 per 1,000 person-years) after COVID-19. There was an increased risk of individual diagnoses of mood disorders including depressive episodes (1.46, 1.29–1.65, $P$ = 9.92 × 10$^{-10}$; 4.14, 2.64–5.83 per 1,000 person-years) and anxiety disorders including panic disorder (1.63, 1.02–2.62, $P$ = 0.041; 0.35, 0.01–0.90 per 1,000 person-years), generalized anxiety disorder (1.63, 1.44–1.84, $P$ = 7.40 × 10$^{-15}$; 5.25, 3.67–7.02 per 1,000 person-years) and post-traumatic stress disorder (2.00, 1.06–3.76, $P$ = 0.031; 0.21, 0.01–0.57 per 1,000 person-years).

*Antipsychotics, antidepressants, benzodiazepines or mood stabilizers.* Coupled with the increased risk of psychiatric disorders, there were increased risks of incident prescriptions for antipsychotics (2.99, 2.11–4.24, $P$ = 7.47 × 10$^{-10}$; 2.26, 1.26–3.67 per 1,000 person-years), antidepressants (1.57, 1.30–1.89, $P$ = 1.92 × 10$^{-6}$; 4.21, 2.25–6.58 per 1,000 person-years), benzodiazepines (1.70, 1.45–1.99, $P$ = 3.17 × 10$^{-11}$; 6.45, 4.18–9.11 per 1,000 person-years) and mood stabilizers (1.41, 1.18–1.67, $P$ < 0.001; 3.28, 1.46–5.45 per 1,000 person-years). The risk of prescriptions for subtypes of antidepressant including selective serotonin reuptake inhibitor (SSRI), serotoninnoradrenaline reuptake inhibitor (SNRI) and others were also increased.

*Opioids.* The risk of incident opioid prescriptions was increased (1.39, 1.24–1.55, $P$ = 1.32 × 10$^{-8}$; 9.10, 5.62–13.00 per 1,000 person-years).

*Substance use disorders.* The risk of incident substance use disorders was only marginally increased. For individual outcomes, there was an increased risk for alcohol use disorder (1.51, 1.16–1.98, $P$ = 0.002; 1.05, 0.32–2.00 per 1,000 person-years) but not for tobacco use disorder.

*Sleep disorders.* The risk of incident sleep disorders was increased (1.47, 1.21–1.80, $P$ < 0.001; 1.47, 0.65–2.46 per 1,000 person-years).

*Composite incident outcomes.* Compared with the contemporary control group, participants in the infection group were at an increased risk of any incident psychiatric diagnosis (1.47, 1.35–1.60, $P$ = 8.82 × 10$^{-20}$; 10.49, 7.88–13.32 per 1,000 person-years), any incident psychotropic prescription (1.51, 1.38–1.65, $P$ = 7.33 × 10$^{-19}$; 20.42, 15.16–26.18 per 1,000 person-years) and any incident mental health outcome (1.54, 1.42–1.67, $P$ = 1.70 × 10$^{-24}$; 27.36, 21.16–34.10 per 1,000 person-years) after COVID-19.

*Composite incident or recurrent outcomes.* Overall, the risks of incident or recurrent psychiatric diagnoses (1.44, 1.36–1.54, $P$ = 8.36 × 10$^{-30}$; 15.07, 12.05–18.28 per 1,000 person-years) and psychotropic prescriptions (1.13, 1.09–1.18, $P$ = 5.13 × 10$^{-10}$; 28.75, 19.29–38.58 per 1,000 person-years) were increased as was the risk of any incident or recurrent mental health outcome (1.19, 1.14–1.23, $P$ = 4.25 × 10$^{-25}$; 44.43, 34.13–55.10 per 1,000 person-years). Figure 4 shows the Kaplan–Meier curves for composite mental health outcomes after SARS-CoV-2 infection.

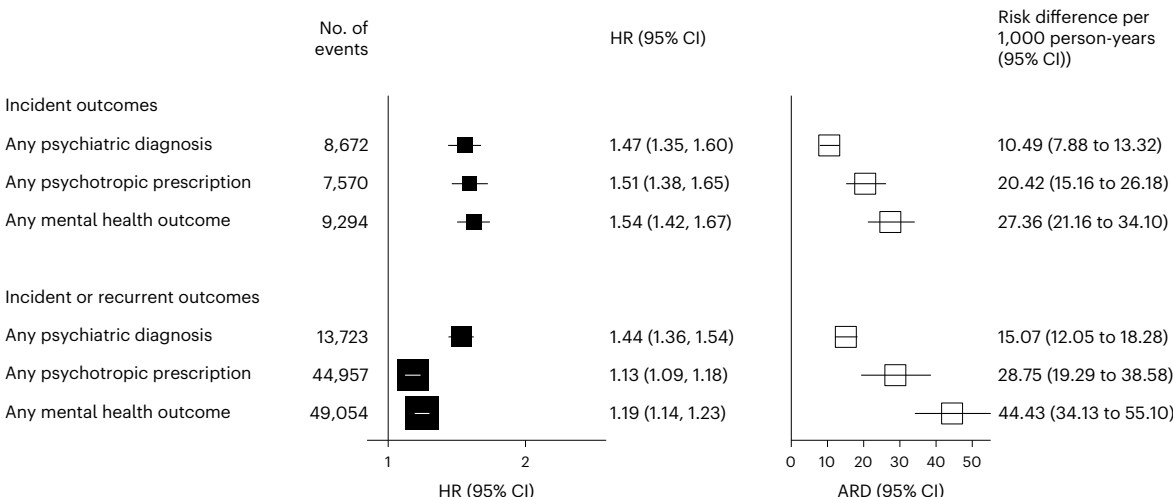

**Fig. 3 | Risks of composite mental health outcomes after SARS-CoV-2 infection compared with the contemporary control group.** Mental health outcomes were ascertained after SARS-CoV-2 infection until the end of follow-up. Results compare the SARS-CoV-2 infection group (*n* = 19,353) to the contemporary control group with no evidence of infection (*n* = 301,398). HRs were adjusted for predefined and data-driven covariates. Risk was reported in relative scale (HRs and 95% CIs) and absolute scale (absolute risk difference per 1,000 person-years with 95 CIs). Squares represent estimates of HRs (with area inversely proportional to the variance) or risk difference, and error bars represent the corresponding 95% CIs.

*Subgroup analyses.* The risks of incident composite mental health outcomes were consistently increased in all subgroups based on age, BMI, IMD, sex, ethnicity and calendar period (Fig. 5).

**COVID-19 group versus contemporary control by vaccination status or test setting.** We further conducted analyses in mutually exclusive groups based on test setting of infection or vaccination status of participants infected with SARS-CoV-2. Among the infection group without a history of mental health outcomes 2 years before follow-up, 2,814 participants were tested positive in a hospital setting and 17,343 were tested positive in a community setting. A total of 13,508 participants were unvaccinated or partially vaccinated, and 6,649 participants were fully vaccinated when tested positive. Assessment of covariate balance after propensity score (PS) weighting suggested that the demographic and medical characteristics of these groups were well balanced. Compared with the contemporary control, the risks of both incident and prevalent mental health outcomes were increased in participants with non-breakthrough infection, and no significant association was observed between breakthrough infection and mental health outcomes, except for an increased risk of psychotropic prescription after breakthrough infection (Fig. 6a). Compared with the contemporary control group, the risks of both incident and prevalent mental health outcomes were increased in those who tested positive in a community setting and were highest in those who tested positive in a hospital setting (Fig. 6b).

**COVID-19 group versus historical control.** After weighting, all characteristics were well balanced between infection and historical control groups (ASMD < 0.1), and the distribution of index dates was fully aligned. The results were consistent with analyses using the contemporary control as the reference group and showed increased risk of mental health outcomes in the infection group compared with historical control group (Supplementary Figs. 4–6).

**COVID-19 group versus test-negative control.** We assessed the risk of mental health outcomes in the infection group compared with the test-negative control group (*n* = 121,563). Characteristics between groups were balanced after weighting. Compared with the test-negative control, the risks of any incident psychiatric diagnosis and mental health outcome were significantly decreased in the infection group.

There was no significant difference in the risk for any incident psychotropic prescription between groups (Supplementary Table 6).

**Positive and negative outcome controls.** Using the same study design and analytical methods, the results of positive outcome controls showed that, compared with the contemporary group, participants in the infection group were at increased risk of established long-COVID symptoms including fatigue and dyspnoea (Supplementary Table 7), which is consistent with previous evidence. The results of negative outcome controls showed that there was no significant association between SARS-CoV-2 infection and subsequent risk of skin neoplasms and skin follicular cysts (Supplementary Table 7).

### Sensitivity analyses
The main results of incident mental health outcomes were robust in multiple sensitivity analyses. Sensitivity analyses using PS weighting, extending the look-back window for the data-driven covariates to 3 years, or excluding participants with a history of outcomes in the past 5 years before follow-up indicated consistent results of increased risk of incident mental health outcomes in the infection group compared with the contemporary control group (Supplementary Table 8). Compared with participants who tested positive in the community setting, those who tested positive in the hospital setting were at increased risk of mental health outcomes (Supplementary Table 8), which is consistent with the main analyses stratified by test setting (Fig. 6b).

### Discussion
In this large-scale prospective community-based cohort, participants with SARS-CoV-2 infection were at increased risks of subsequent incident psychiatric diagnoses (including psychotic disorders, mood disorders, anxiety disorders, alcohol use disorders and sleep disorders), as well as related prescriptions for psychotropic medications (including antipsychotics, antidepressants, benzodiazepines, mood stabilizers and opioids) compared with participants with no evidence of infection in the contemporary control group who experienced similar social and environmental stressors related to the pandemic. The results were consistent when comparing the SARS-CoV-2 infection group with the historical control group that predated the pandemic. These risks were evident even in those who tested positive in the community setting, likely consisting of infected people with mild symptoms or

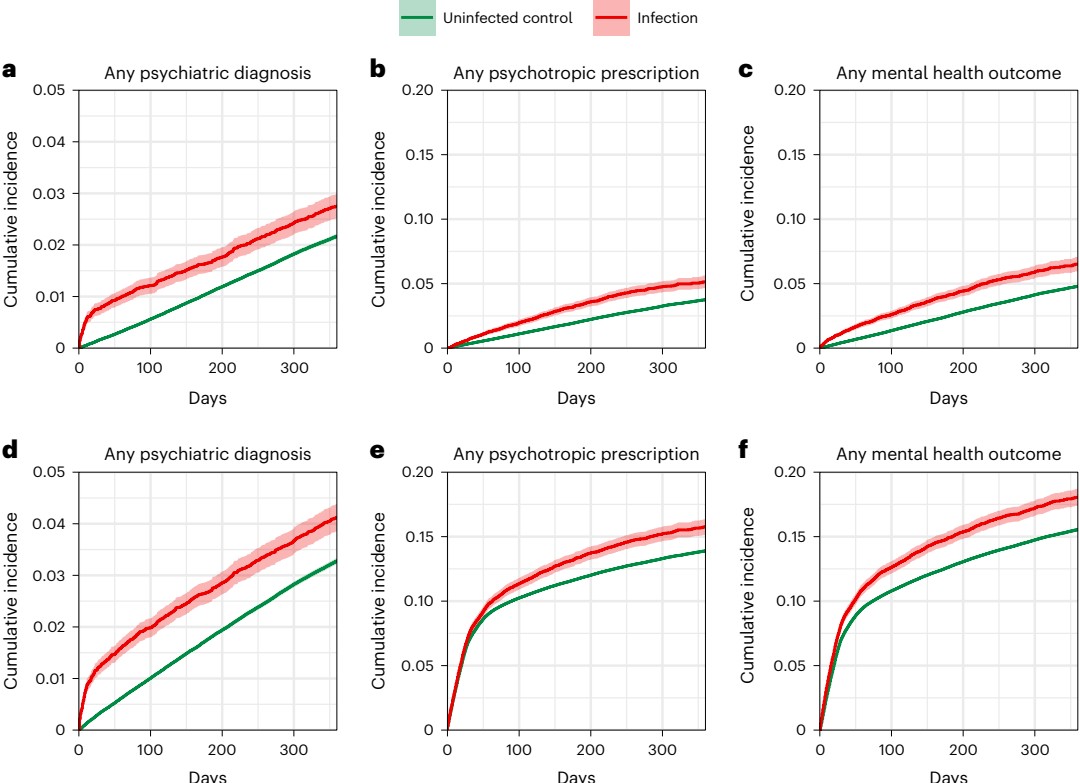

**Fig. 4 | Cumulative incidence curves of mental health outcomes after SARS-CoV-2 infection compared with the contemporary control group.** Mental health outcomes were ascertained after SARS-CoV-2 infection until the end of follow-up. Results compare the SARS-CoV-2 infection group ($n$ = 19,353) to the contemporary control group with no evidence of infection ($n$ = 301,398). HRs were adjusted for predefined and data-driven covariates. The shading around plotted lines shows 95% CIs. **a–c**, incident outcome; **d–f**, incident or recurrent outcome.

asymptomatic COVID-19, and were highest in those who tested positive in the hospital and might have had more severe COVID-19. Fully vaccinated participants with breakthrough infection were at lower risk of mental health outcomes compared with those who were unvaccinated or partially vaccinated when they got infected. The increased risk of mental health outcomes was robust in multiple sensitivity analyses. Overall, these findings suggest that survivors of COVID-19 are at increased risk of subsequent psychiatric disorders and related psychotropic prescriptions at 1 year. Vaccination may potentially have additional benefits of alleviating long-term psychiatric sequelae of COVID-19 beyond protecting against COVID-19 infection and severe complications.

Previous studies based on EHR data suggested that individuals with or without a history of mental illness had an increased risk of psychiatric conditions in the subsequent 3–12 months following acute SARS-CoV-2 infection compared with individuals without evidence of the infection or individuals with other acute respiratory infections such as influenza[5–8]. Nonetheless, several important socioeconomic and lifestyle factors such as physical activity and alcohol drinking associated with both SARS-CoV-2 infection and mental health outcomes were largely unavailable in these studies, possibly leading to residual confounding. There may also be recording or surveillance bias due to restrictions and disruptions in patient help-seeking behaviours during the the pandemic. EHR data are also more vulnerable to recording or surveillance bias and the potential impact of the disruptions in healthcare services during the early period of the pandemic compared with the prospective cohort based on the general population. In addition, recruitment in the EHR study is dependent on patients' utilization of the health system, which may vary substantially across countries with different healthcare delivery systems and limits the generalizability of findings beyond the population covered by the respective EHR

database. For example, a large US study of psychiatric sequelae based on EHR data from discharged veterans, 90% of whom were male, was unlikely to be representative of the general population[8]. The estimates of absolute risk difference or disease burden is particularly susceptible to the selected samples due to the difference in their baseline risk. Using a large prospective cohort recruited before the pandemic in the UK, our study provides more precise, representative relative and absolute risk estimates that corroborate previous EHR-based reports suggesting an increased risk of incident psychiatric disorders and psychotropic prescriptions after SARS-CoV-2 infection.

Our findings on the risk of psychiatric sequelae after breakthrough infection also address the existing knowledge gap and highlight that vaccination (independent of vaccine type) potentially has additional benefits of alleviating long-term psychiatric sequelae of COVID-19 beyond protecting against SARS-CoV-2 infection and severe complications. Although previous US-based EHR studies assessed the impact of vaccination before infection on the risk of several mental health outcomes, such as psychotic disorder and anxiety, during a maximum of 6 months' follow-up[10,11], the effect of vaccination on the full spectrum of mental health outcomes over longer follow-up periods remains unclear. Our study demonstrated that full vaccination before infection may significantly reduce the psychiatric sequelae of COVID-19 as proxied by psychiatric diagnosis or psychotropic prescription during the 1 year of follow-up. Our findings are also consistent with previous observations of reduced long-COVID symptoms (including non-specific mental health symptoms such as psychological distress and frailty) after full vaccination. A recent study in the UK suggested that long-covid symptoms such as trouble sleeping, worry and weakness were observed to decrease after vaccination and there was sustained improvement after two doses of vaccination[12]. Another UK-based study suggested that fully vaccinated individuals with breakthrough

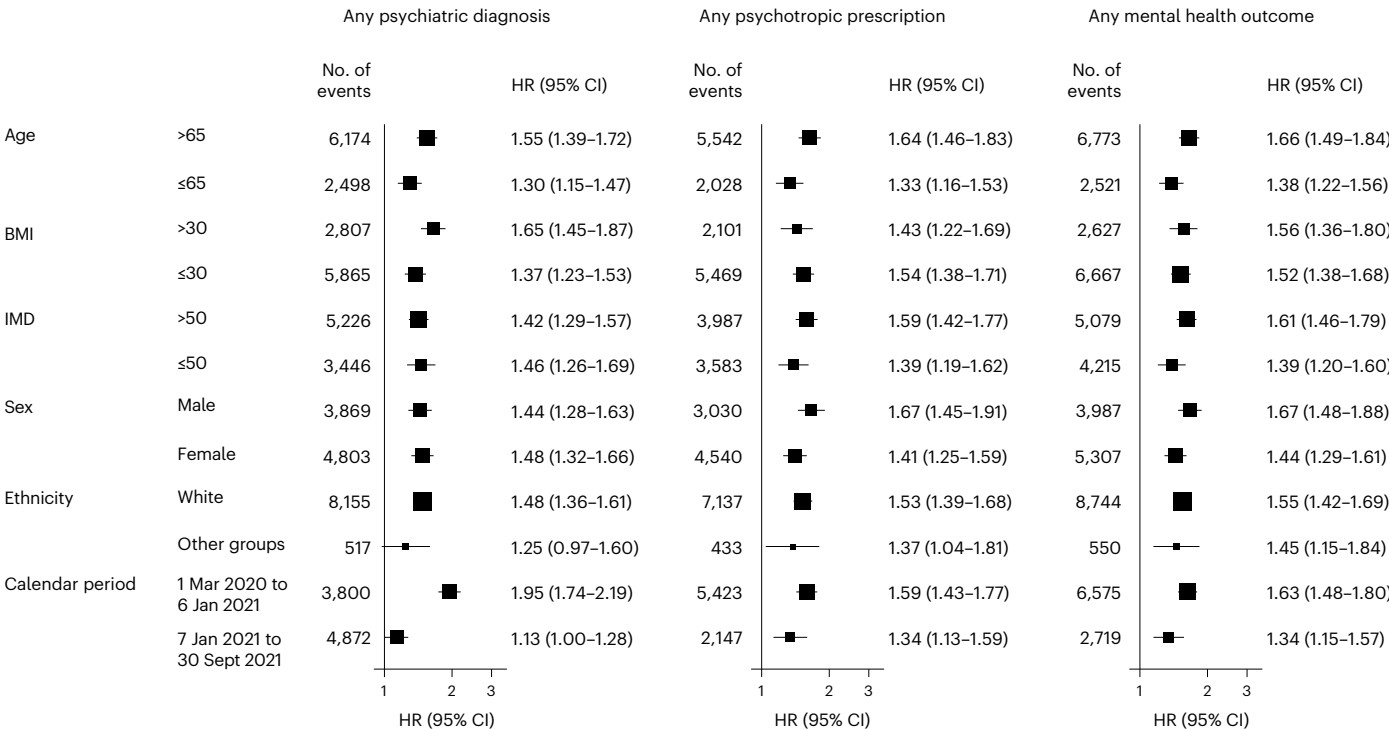

| | | Any psychiatric diagnosis | | Any psychotropic prescription | | Any mental health outcome | |
|---|---|---|---|---|---|---|---|
| | | No. of events | HR (95% CI) | No. of events | HR (95% CI) | No. of events | HR (95% CI) |
| Age | >65 | 6,174 | 1.55 (1.39–1.72) | 5,542 | 1.64 (1.46–1.83) | 6,773 | 1.66 (1.49–1.84) |
| | ≤65 | 2,498 | 1.30 (1.15–1.47) | 2,028 | 1.33 (1.16–1.53) | 2,521 | 1.38 (1.22–1.56) |
| BMI | >30 | 2,807 | 1.65 (1.45–1.87) | 2,101 | 1.43 (1.22–1.69) | 2,627 | 1.56 (1.36–1.80) |
| | ≤30 | 5,865 | 1.37 (1.23–1.53) | 5,469 | 1.54 (1.38–1.71) | 6,667 | 1.52 (1.38–1.68) |
| IMD | >50 | 5,226 | 1.42 (1.29–1.57) | 3,987 | 1.59 (1.42–1.77) | 5,079 | 1.61 (1.46–1.79) |
| | ≤50 | 3,446 | 1.46 (1.26–1.69) | 3,583 | 1.39 (1.19–1.62) | 4,215 | 1.39 (1.20–1.60) |
| Sex | Male | 3,869 | 1.44 (1.28–1.63) | 3,030 | 1.67 (1.45–1.91) | 3,987 | 1.67 (1.48–1.88) |
| | Female | 4,803 | 1.48 (1.32–1.66) | 4,540 | 1.41 (1.25–1.59) | 5,307 | 1.44 (1.29–1.61) |
| Ethnicity | White | 8,155 | 1.48 (1.36–1.61) | 7,137 | 1.53 (1.39–1.68) | 8,744 | 1.55 (1.42–1.69) |
| | Other groups | 517 | 1.25 (0.97–1.60) | 433 | 1.37 (1.04–1.81) | 550 | 1.45 (1.15–1.84) |
| Calendar period | 1 Mar 2020 to 6 Jan 2021 | 3,800 | 1.95 (1.74–2.19) | 5,423 | 1.59 (1.43–1.77) | 6,575 | 1.63 (1.48–1.80) |
| | 7 Jan 2021 to 30 Sept 2021 | 4,872 | 1.13 (1.00–1.28) | 2,147 | 1.34 (1.13–1.59) | 2,719 | 1.34 (1.15–1.57) |

**Fig. 5 | Subgroup analyses of the risks of incident mental health outcomes after SARS-CoV-2 infection compared with the contemporary control group.** Mental health outcomes were ascertained after SARS-CoV-2 infection until the end of follow-up. Results compare the SARS-CoV-2 infection group (*n* = 19,353) to the contemporary control group with no evidence of infection (*n* = 301,398) by predefined subgroups. HRs were adjusted for predefined and data-driven covariates where applicable. The calendar period was stratified according to the timeline of UK government coronavirus lockdowns and measures when England entered its third national lockdown on 6 January 2021.

infections had significantly lower risk of long-duration symptoms (≥28 days; odds ratio: 0.51, 95% CI 0.32–0.82)[13]. The putative mechanism underlying the protective effect is that the reduced severity of infection after full vaccination may then translate into a lower risk of long-term psychiatric sequelae[10], which was supported by the higher risk of sequelae in those with more severe disease (treated in the hospital setting). Given the existing large number of COVID-19 survivors (so far, ~700 million globally) and the increasing infections worldwide accompanying the loosening of COVID-19 restrictions, the absolute risk of incident psychiatric disorders may translate into an enormous global burden of mental health. Suppose the benefits of vaccination on long-term psychiatric manifestations of COVID-19 were confirmed by independent prospective studies, the vaccine should be considered as part of public health strategies against the long-term symptoms of COVID-19, given the substantial medical costs associated with treating these related mental disorders. As vaccination may only partially reduce the risk of long-term psychiatric sequelae, policymakers and health systems should also develop priorities and long-term strategies for the early identification and treatment of affected individuals to mitigate psychiatric sequelae and enhance wellbeing especially in vulnerable survivors of COVID-19.

Notably, we found no significant difference in the risk of psychiatric diagnosis at 1 year after breakthrough infection compared with the contemporary vaccinated control, while the risk of psychotropic prescription was marginally increased. Previous evidence showed that participants with breakthrough infection, compared with contemporary controls, were still at higher risk of mental health outcomes during a short-term follow-up (≤6 months)[10]. Our long-term results suggest that the risk of clinically diagnosed psychiatric disorders after breakthrough infection was similar to that of vaccinated participants with no evidence of infection during the extended 1-year follow-up period. However, the slight increase in psychotropic prescription in

primary care, where non-specific symptoms of mental health problems such as psychological distress (for example, symptoms of stress, depression and anxiety) are likely to be the main driver, indirectly suggests that the excess burden of subclinical mental health problems related to breakthrough infection may still exist at 1 year. In view of the expanding scale of the pandemic and the global vaccination campaign, cases of breakthrough infection may continue to accumulate. Preventive measures and psychological interventions at the population level should be informed to prevent and alleviate the long-term consequences of breakthrough infection on subclinical mental health problems, and to promote resilience and help prevent escalation to severe mental disorders.

Comparing test-positive with test-negative individuals, we found no increased risk of mental health outcomes at 1 year. These results are consistent with a Danish registry study suggesting that the risks of mental health outcomes (depression, anxiety disorders and psychosis) were not different or even decreased in the infection group compared with the test-negative group[14]. A UK primary care registry study also found that the risk of psychiatric morbidity in individuals with negative SARS-CoV-2 test results was increased compared with the general population with no evidence of infection, and this association was similar to that observed in test-positive individuals[15]. Similarly, a recent nationwide EHR study from Israel suggested that compared with participants with a negative PCR test, the risk of depression and anxiety were not differential and the risk of psychosis was even decreased in those with a positive test during the long-term follow-up (180–360 days after infection)[16]. Although having a negative test result should not directly affect mental health outcomes, the testing behaviour in the circumstance of underlying non-infection is likely to indicate that potential occupational and behavioural factors predict a higher risk of subsequent mental health outcomes. For example, healthcare workers who require more frequent COVID-19 tests even without any symptoms

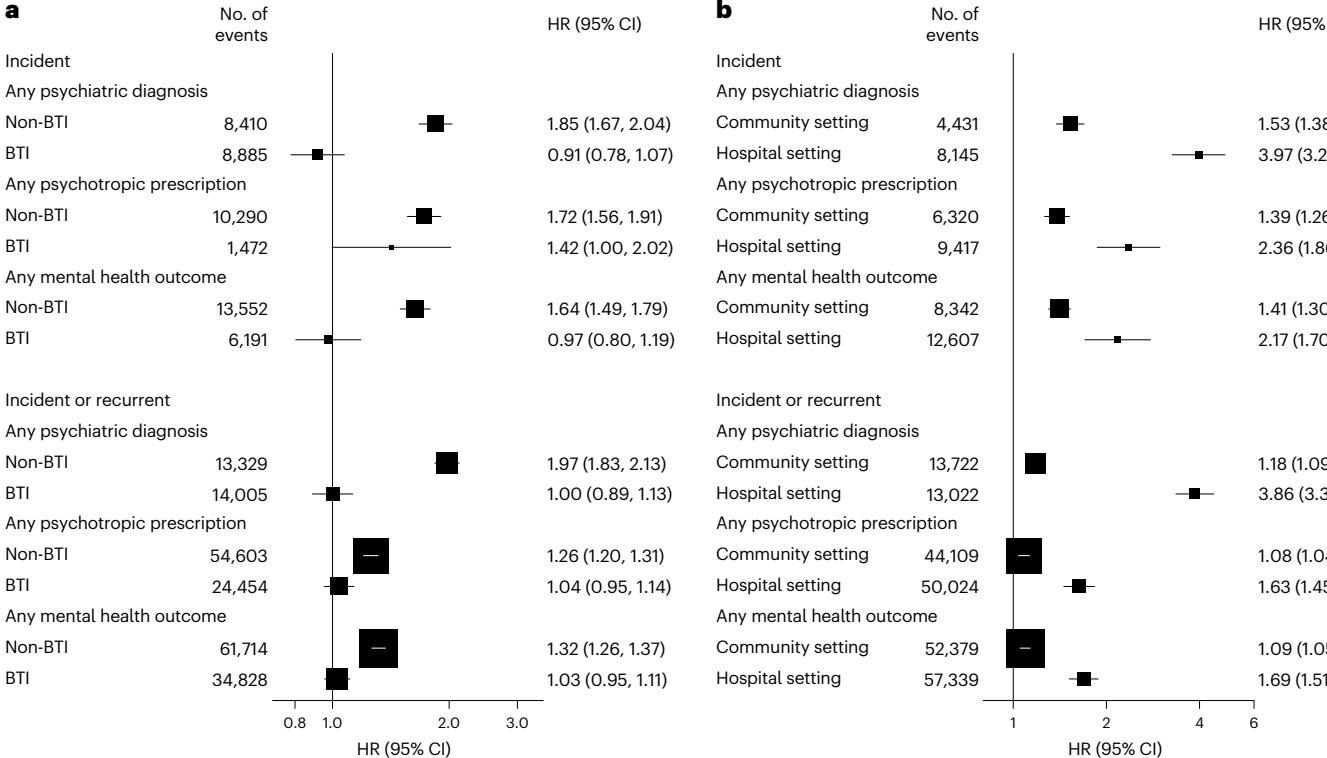

**Fig. 6 | Risks of psychiatric diagnoses and psychotropic prescriptions by vaccination status and test setting of participants in the SARS-CoV-2 infection group compared with the contemporary control group.** Mental health outcomes were ascertained after SARS-CoV-2 infection until the end of follow-up. HRs were adjusted for predefined and data-driven covariates. **a**, Analyses by vaccination status. To account for health behaviours related to vaccines, the breakthrough infection (BTI) group (n = 6,453) was compared to the vaccinated control group (n = 117,540), and the non-BTI group (n = 12,900) to the unvaccinated or partially vaccinated control group (n = 183,858). **b**, Analyses by test setting. Results compare the SARS-CoV-2 infection group by test setting (hospital setting n = 2,541; community setting n = 16,812) to the contemporary control group with no evidence of infection (n = 301,398).

may experience excess psychological distress and be vulnerable to mental illness[17,18]. In addition, individuals who seek a test could be experiencing health anxiety and are predisposed to mental health issues. A previous study has also suggested that individuals with negative test results had a higher proportion of previous mental health disorders than those with positive results[15]. Nevertheless, previous large US registry studies[6–8] supporting an increased risk of mental health outcomes after COVID-19 did not specifically compare and report the risk between the test-positive and test-negative groups, and this finding should be interpreted with caution and warrants further research. An alternative explanation to our findings could be that testing behaviour rather than infection could at least partially account for the observed increase in mental health outcomes after COVID-19. However, our observations of a further increase in risk among those with severe (hospitalized) COVID-19 and in the unvaccinated or partially unvaccinated would support a causal association between SARS-CoV-2 infection and subsequent mental health outcomes.

COVID-19 may impact subsequent mental health directly and indirectly through several plausible mechanisms at both biological and environmental levels. Stress-evoking and disruptive societal changes during the pandemic may indirectly have detrimental impact on mental health in the general population[19]. Specifically, the COVID-19 pandemic and related public health interventions such as quarantine and social distancing may have long-term adverse mental health consequences, especially on vulnerable groups including patients diagnosed with COVID-19 and those with pre-existing mental disorders[18,20]. These disproportional impacts may partly explain the increased risk of mental health outcomes after infection compared with the contemporary control, although participants in both groups experienced similar pandemic-related socioeconomic and environmental stressors.

In addition, possible changes in behaviours such as decreased physical activity, having a poor diet, and increased avoidance of health care and social contact in some individuals following recovery from acute COVID-19 may also contribute to the increased risk of long-term psychiatric sequelae[2,20]. Overlapping biological factors between viral infection and psychiatric disorders may also be implicated. Several possible underlying mechanisms include increased blood–brain barrier permeability and the central nervous system infiltration of SARS-CoV-2, chronic systemic immuno-inflammatory responses, dysregulation of microglia and astrocytes, and disturbances in synaptic signalling of upper-layer excitatory neurons[21,22]. Future studies are needed to explore whether post-COVID-19 psychiatric disorders result from SARS-CoV-2 infection itself, the disproportional adverse effects of pandemic-related factors or a combination of both.

The current study systematically explored the long-term psychiatric sequelae of COVID-19 in a prospective cohort based on the general population with comprehensive and reliable recorded data. The potential benefits of vaccination on psychiatric sequelae reinforce the need for vaccination and support the ongoing global vaccination campaigns. Overall, the findings are robust given the large sample size, the use of PS weighting, and the consistent results in sensitivity and secondary analyses. However, the findings from this study should be interpreted with caution in the context of its limitations. First, concerns have been raised that participants in the UK Biobank may be suboptimally representative of the whole population in the UK and were likely to be older and generally healthier. These issues primarily affect the estimates of absolute incidence rates[23]. Although the relative risk between comparison groups was largely not affected, this might still limit the generalizability of our findings to younger populations. Second, although we used robust statistical approaches

such as PS weighting and PS matching based on a set of covariates to adjust for potential differences in characteristics between comparison groups, residual confounding cannot be ruled out in this observational study. Mendelian randomization is less susceptible to potential biases that are common in conventional observational studies. However, the heritability of 7 single-nucleotide polymorphisms, identified in the current largest genome-wide association study of SARS-CoV-2 infection, that reached genome-wide significance was low at 0.17%[24], which may lead to weak instrument bias and preclude the conduct of valid Mendelian randomization analyses at the current stage. Future Mendelian randomization studies are needed to further clarify whether the observed association is causal when robust genetic variants associated with COVID-19 are available. Third, a proportion of participants in the contemporary control group may have undiagnosed or untested COVID-19. However, this would tend to underestimate the risk estimates and thus lead to more conservative results. The linkage of UK Biobank participants to official national databases for COVID-19 testing and hospitalization meant that the likelihood of misclassification of infected and uninfected participants was minimized. Fourth, we did not statistically correct for multiple comparisons, although most results were significant with a $P$ value less than 0.0001. Fifth, although we observed a significantly increased risk of a series of psychiatric diagnoses, the case number of several disorders was relatively small, limiting further analysis of subcategories, especially severe ones such as schizophrenia. Finally, the risk estimates from our analyses may be representative of the mixed effect of several SARS-CoV-2 strains (the alpha and delta variants were dominant during different periods of follow-up), which should be cautiously extrapolated to novel variants, such as omicron. However, one recent study found that the risks of neurological and psychiatric outcomes after the emergence of the omicron (B.1.1.529) variant were similar to those after the delta (B.1.617.2) variant[25]. Another study also suggested that the wild-type, alpha and delta variants resulted in similar long-term COVID-19 sequelae[16]. The epidemiology of COVID-19 psychiatric sequelae may also change with the evolving pandemic, emerging variants and increasing vaccine uptake, and further studies are warranted.

In this large-scale prospective cohort study, people who survived the acute phase of COVID-19 were at increased risk of subsequent incident psychiatric disorders and psychotropic prescriptions. These risks were higher in those with more severe disease, treated in hospital settings, and were significantly reduced in fully vaccinated people. Future independent studies are needed to verify the potential benefits of the vaccine on the psychiatric sequelae of COVID-19 and to inform other approaches to enhance mental wellbeing. Identification and treatment of psychiatric disorders among survivors of SARS-CoV-2 infection should be a priority in the long-term management of COVID-19, especially for those with severe infection and those who were not fully vaccinated at the time of infection.

## Methods

### Study design and participants

We used data from UK Biobank[26] (https://www.ukbiobank.ac.uk/) to conduct this study. All participants provided written informed consent at the time of recruitment. This study followed the Strengthening the Reporting of Observational Studies in Epidemiology (STROBE) reporting guidelines[27] and received ethical approval from the UK Biobank Ethics Advisory Committee (application 65397).

The UK Biobank is an ongoing community-based prospective cohort study, which recruited more than 500,000 participants out of 9.2 million adults aged 40–70 years in the UK who were invited to participate (5.5% response rate), as detailed elsewhere[26]. The baseline survey took place from 2006 to 2010 in 22 assessment centres. Overall, 503,317 participants provided written informed consent to take part in the study and be followed up through linkage to health-related records. PCR-based testing results for SARS-CoV-2 were obtained from Public

Health England's Second Generation Surveillance System (PHE-SGSS), a centralized microbiology database covering English clinical diagnostics laboratories that had been previously validated for COVID-19 research[28]. Records of psychiatric diagnoses and relevant medications were obtained by linkage to the Hospital Episode Statistics (HES) and the primary care prescription database, respectively, and relevant confounding factors are available in UK Biobank. Potential methodological limitations of EHR studies on long-COVID compared with cohorts based on the general population are provided in Supplementary Table 9.

### Study cohort

We included UK Biobank participants from England who were still alive by 1 March 2020 (date of the first recorded COVID-19 case in the UK Biobank) to construct the SARS-CoV-2 infection group (hereafter referred to as the infection group). The infection group was defined as all individuals who had a positive result on a PCR test for SARS-CoV-2 confirmed between 1 March 2020 and 30 September 2021. The date when the first positive specimen sample was taken was set as the start of follow-up ($T_0$) for the infection group. The non-infected contemporary control group included individuals with no evidence of SARS-CoV-2 infection (those not in the infected group who had negative testing results or were never tested). To ensure that the contemporary control group had a similar follow-up period as the infection group, a random index date was assigned to the contemporary control group during the same observation period (between 1 March 2020 and 30 September 2021) on the basis of the distribution of $T_0$ in the infection group, so that the proportion of participants followed up from a certain date was the same in both comparison groups. The end of follow-up for both the infection and the contemporary control groups was 30 September 2022, with the maximum follow-up period limited to 1 year.

To further examine the associations between SARS-CoV-2 infection and mental health outcomes in comparison to those unaffected by the COVID-19 pandemic, a historical control group was constructed by including participants from UK Biobank who were alive by 1 March 2017 and were not in the infection group. Similarly, the start of follow-up for participants in the historical control group was randomly assigned according to the distribution of $T_0$ in the infection group as $T_0$ minus 3 years (1,095 days). The end of the follow-up period for the historical control group was 30 September 2019, with the maximum follow-up period limited to 1 year.

To provide additional benchmarking for the incidence and risk of mental health outcomes, we constructed an additional control group including participants with negative SARS-CoV-2 test results. Participants who tested negative for SARS-CoV-2 between 1 March 2020 and 30 September 2021, and were not in the infection group were included in the test-negative group. Follow-up time of the test-negative group was assigned to match the distribution of follow-up time in the infection group. The end of follow-up for the test-negative group was 30 September 2022, with the maximum follow-up period limited to 1 year.

These cohorts were followed longitudinally to assess the incidence and risk of incident or prevalent (incident or recurrent) psychiatric disorders and prescriptions for psychotropic medications during a maximum of 12 months of follow-up after SARS-CoV-2 infection.

### Mental health outcomes

The mental health-related outcomes including psychiatric disorders and prescriptions for psychotropic medications were predefined on the basis of previous knowledge and previous studies of COVID-19 psychiatric sequelae[5–8]. Psychiatric disorders were diagnosed on the basis of ICD-10 codes (International Cassification of Diseases, 10th revision), including psychotic disorders (F20–F29), mood disorders (F30–F39), anxiety disorders (F40–F48, including trauma-related and somatoform disorders), substance use disorders (F12–F19) and sleep disorders (F51 and G47). We also investigated the major individual outcomes in each category separately. For example, the major components of mood

disorders considered in this study included depressive episode (F32) and mania/bipolar affective disorder (F30 and F31). Prescriptions for psychotropic medications were recorded in the UK Biobank database, including antidepressants, antipsychotics, benzodiazepines, mood stabilizers and opioids. Detailed definitions of mental health outcomes are provided in Supplementary Table 1. We specified three composite outcomes of any psychiatric disorder (F20–F48), any psychotropic prescription and any mental health outcome.

Because psychiatric disorders tend to recur or relapse, we separately estimated the risk of the incident mental health outcomes (for example, excluding participants with a history of the corresponding psychiatric disorders or psychotropic prescriptions in 2 year before the start of follow-up) and the risk of incident or recurrent (prevalent) mental health outcomes (for example, including participants who had a diagnosis or record of related outcomes before the start of follow-up). The incident mental health outcomes after SARS-CoV-2 infection were reported as the primary outcomes. Analyses of diagnostic subcategories and incident or recurrent mental health outcomes are provided in Supplementary Information.

### Covariates

We used both predefined and data-driven covariates to adjust for the difference in baseline characteristics between comparison groups. Predefined covariates were selected on the basis of previous knowledge, including a comprehensive set of established and suspected risk factors for COVID-19 and mental health conditions: age, sex, ethnicity, index of multiple deprivation[29] (a summary contextual measure including seven aspects in crime, education, employment, health, housing, income and living environment used to represent socioeconomic status), smoking status, physical activity, body mass index, systolic and diastolic blood pressure, estimated glomerular filtration rate and hospital admissions. The battery of predefined covariates also included comorbidities identified using all clinical components of the Charlson comorbidity index[30]: cancer, cerebrovascular disease, chronic kidney disease, chronic obstructive pulmonary disease, congestive heart failure, dementia, diabetes, human immunodeficiency virus (HIV)/AIDS, hemiplegia, myocardial infarction, liver disease, renal disease, peripheral vascular disease and peptic ulcer disease.

To further reduce the risk of residual confounding and optimize adjustment of potential confounders, we included a list of data-driven clinical episodes diagnosed during patient hospitalization within 1 year before $T_0$. We first classified 8,651 source ICD-10 diagnosis codes into 453 disease phenotype groups (DPGs) using a validated mapping algorithm (Phecode v.1.2 ICD-10 map)[31]. We further selected DPGs that occurred in more than 0.1% of participants into the adjustment after excluding rare DPGs that can hardly characterize a cohort and may lead to inconsistency in model estimation[32,33]. Of note, when comparing with the historical control group, we additionally adjusted for the level of healthcare utilization during follow-up for the comparison groups, given the substantial disruptions and backlogs in medical care during the pandemic[8,34].

### Statistical analyses

In the main analyses, we used PS weighting to control for differences in baseline characteristics between comparison groups (infection, contemporary control and historical control). For each comparison pair, we built a multivariable logistic regression with Lasso L1 penalty to estimate the PS as the probability of belonging to the exposure (infection) group and the probability of belonging to the control group, using both predefined and data-driven DPGs. Inverse probability weights were calculated as one divided by the PS in the infection group and divided by one minus PS in the control group. We also used PS matching as an alternative analytic approach in sensitivity analysis to verify the robustness of the results from PS weighting. Infected participants were matched 1:10 to the uninfected participants, with a

caliper distance of 0.2 standard error of the logit of the PS and exact matching for $T_0$. Any baseline characteristic with an ASMD between comparison groups lower than 0.1 was considered well balanced[35]. We used cause-specific Cox proportional hazards regression models where death was considered as a competing risk to estimate HRs of mental health conditions between the infection and contemporary groups, and between the infection and historical groups, with the inverse probability weights applied when PS weighting was used. We also estimated the adjusted incidence rate per 1,000 person-years in the infection and control groups, and the difference in incidence rate between comparison groups.

Regarding the risk of incident composite mental health outcomes compared with the contemporary control, we conducted subgroup analyses on the basis of age, sex, BMI, IMD and ethnicity. To assess whether contextual factors of COVID-19, such as quarantine measures, may increase the risk of psychiatric sequelae, we undertook subgroup analyses by calendar period according to the timeline of UK government coronavirus lockdowns and measures (first stage: between January 2020 and January 2021; second stage: between January 2021 and September 2021). The first two national lockdowns came into force in England in the first stage, and England entered its third national lockdown on 6 January 2021[36].

To investigate whether the incidence and risk of mental health outcomes after SARS-CoV-2 infection were affected by vaccination status (independent of vaccine type) and the test setting for COVID-19, we further categorized the SARS-CoV-2 infection group as infection tested in a hospital setting and in a community setting, and categorized the infection group as non-breakthrough infection (unvaccinated or partially 1-dose vaccinated at $T_0$) and breakthrough infection (fully 2-dose vaccinated at $T_0$). We compared these infection subgroups with control groups. Considering the potential impact of health behaviours related to SARS-CoV-2 vaccines, we especially compared the breakthrough infection group with vaccinated control group (being vaccinated at assigned $T_0$ but without subsequent infection), and compared the non-breakthrough infection group with the unvaccinated or partially vaccinated control group.

To provide additional benchmarking for the incidence and risk of composite mental health outcomes, we additionally compared the infection group with the control group of participants with negative SARS-CoV-2 test results. Comparisons were conducted using cause-specific Cox proportional hazards regression models balancing by weighting using the same set of covariates.

To assess the robustness of our main results on incident composite mental health outcomes, we conducted multiple sensitivity analyses. First, we used the 1:10 PS matching approach to construct contemporary and historical control groups with comparable characteristics. Second, we extended the look-back window for the data-driven clinical variables to 3 years. Third, in addition to the analysis stratified by test setting, we directly compared the risk of mental health outcomes between hospitalized and non-hospitalized COVID-19 patients. Finally, we excluded participants with a history of the mental health outcomes in the past 5 years before the start of follow-up and repeated the analyses of incident outcomes.

### Positive and negative controls

To assess whether our study design and analytical approach could replicate established knowledge, we investigated the association between SARS-CoV-2 infection and the risk of fatigue and dyspnoea (common long-COVID symptoms defined by the WHO) as a positive outcome control where positive associations are expected.

We then examined the association between SARS-CoV-2 infection and a set of two negative outcome controls (skin neoplasms and skin follicular cysts) where no association is expected based on previous evidence. The successful use of both positive and negative outcome controls can reduce concerns about unaccounted biases related to

study design, analytical methods, residual confounding and other latent sources of bias[37].

Statistical significance was determined using a 95% CI that excluded 1 for ratios and 0 for rate differences. All analyses and data visualizations were conducted using R (v.4.1).

## Reporting summary

Further information on research design is available in the Nature Portfolio Reporting Summary linked to this article.

## Data availability

Researchers can apply to use the UK Biobank dataset by registering and applying at https://ukbiobank.ac.uk/register-apply/. Any additional summary data generated and/or analysed during the current study are available from the corresponding author on reasonable request.

## Code availability

Analysis code is available at https://github.com/xjq8065524/COVID_Mental_Health.

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

## Acknowledgements

Y.W. was funded by the Clarendon Fund Scholarship. J.X. was funded by the Jardine-Oxford Graduate Scholarship and a titular Clarendon Fund Scholarship. C.G.-R. was funded by the Carlos III Health Institute (ISCIII) and co-financed by the European Union (PI20/00661). D.P.-A. was funded by an NIHR Senior Research Fellowship (grant SRF-2018-11-ST2-004). The research was funded/supported by the National Institute for Health Research (NIHR) Oxford Biomedical Research Centre (BRC). The views expressed are those of the authors and not necessarily those of the NHS, the NIHR or the Department of Health. The funding organizations had no role in the design and conduct of the study; collection, management, analysis and interpretation of the data; preparation, review or approval of the manuscript; and decision to submit the manuscript for publication. We thank M. V. Vitiello for reviewing the manuscript.

## Author contributions

Y.W. and J.X. had full access to all the data in the study and take responsibility for the integrity of the data and the accuracy of the data analyses. Y.W., J.X. and D.P.-A. conceptualized and designed the study, and acquired, analysed or interpreted data. Y.W. drafted the manuscript. All authors critically revised the manuscript for important intellectual content. J.X. conducted statistical analysis. D.P.-A. obtained funding, provided administrative, technical or material support, and supervised the project.

## Competing interests

C.G.-R. has received honoraria/travel support from Abbot, Angelini, Cassen-Recordati, Janssen-Cilag, and Lundbeck. D.P.-A. reported grants from Amgen, UCB Biopharma, Les Laboratoires Servier, Novartis, and Chiesi-Taylor, as well as speaker fees and advisory board membership with AstraZeneca and Johnson and Johnson outside the submitted work, in addition to research support from Janssen. The remaining authors declare no competing interests.

## Additional information

**Correspondence and requests for materials** should be addressed to Junqing Xie.

¹Nuffield Department of Population Health, University of Oxford, Oxford, UK. ²School of Population Medicine and Public Health, Chinese Academy of Medical Sciences/Peking Union Medical College, Beijing, China. ³Centre for Statistics in Medicine and NIHR Biomedical Research Centre Oxford, NDORMS, University of Oxford, Oxford, UK. ⁴Key Laboratory of Aging-related Bone and Joint Diseases Prevention and Treatment, Ministry of Education, Xiangya Hospital, Central South University, Changsha, China. ⁵Department of Orthopaedics, Xiangya Hospital, Central South University, Changsha, China. ⁶Barcelona Clinic Schizophrenia Unit, Hospital Clínic de Barcelona, Departament de Medicina, Institut de Neurociències (UBNeuro), Universitat de Barcelona (UB), Barcelona, Spain. ⁷CIBERSAM, ISCIII, Madrid, Spain. ⁸Institut d'Investigacions Biomèdiques August Pi I Sunyer (IDIBAPS), Barcelona, Spain. ⁹Medical Informatics, Erasmus Medical Center University, Rotterdam, the Netherlands. ¹⁰These authors contributed equally: Yunhe Wang, Binbin Su. ✉e-mail: Junqing.xie@ndorms.ox.ac.uk

# Reporting Summary

## Statistics

For all statistical analyses, confirm that the following items are present in the figure legend, table legend, main text, or Methods section.

| n/a | Confirmed | |
|---|---|---|
| ☐ | ☒ | The exact sample size (*n*) for each experimental group/condition, given as a discrete number and unit of measurement |
| ☒ | ☐ | A statement on whether measurements were taken from distinct samples or whether the same sample was measured repeatedly |
| ☐ | ☒ | The statistical test(s) used AND whether they are one- or two-sided<br>*Only common tests should be described solely by name; describe more complex techniques in the Methods section.* |
| ☐ | ☒ | A description of all covariates tested |
| ☐ | ☒ | A description of any assumptions or corrections, such as tests of normality and adjustment for multiple comparisons |
| ☐ | ☒ | A full description of the statistical parameters including central tendency (e.g. means) or other basic estimates (e.g. regression coefficient) AND variation (e.g. standard deviation) or associated estimates of uncertainty (e.g. confidence intervals) |
| ☐ | ☒ | For null hypothesis testing, the test statistic (e.g. $F$, $t$, $r$) with confidence intervals, effect sizes, degrees of freedom and $P$ value noted<br>*Give P values as exact values whenever suitable.* |
| ☒ | ☐ | For Bayesian analysis, information on the choice of priors and Markov chain Monte Carlo settings |
| ☐ | ☒ | For hierarchical and complex designs, identification of the appropriate level for tests and full reporting of outcomes |
| ☐ | ☒ | Estimates of effect sizes (e.g. Cohen's *d*, Pearson's *r*), indicating how they were calculated |

*Our web collection on statistics for biologists contains articles on many of the points above.*

## Software and code

Policy information about availability of computer code

| | |
|---|---|
| Data collection | No software was used. |
| Data analysis | All analyses and data visualizations were conducted using R statistical software (version 4.1). |

For manuscripts utilizing custom algorithms or software that are central to the research but not yet described in published literature, software must be made available to editors and reviewers. We strongly encourage code deposition in a community repository (e.g. GitHub). See the Nature Portfolio guidelines for submitting code & software for further information.

## Data

Policy information about availability of data

All manuscripts must include a data availability statement. This statement should provide the following information, where applicable:
- Accession codes, unique identifiers, or web links for publicly available datasets
- A description of any restrictions on data availability
- For clinical datasets or third party data, please ensure that the statement adheres to our policy

Researchers can apply to use the UK Biobank dataset by registering and applying at https://ukbiobank.ac.uk/register-apply/. Any additional summary data generated and/or analyzed during the current study are available from the corresponding author on reasonable request.

# Human research participants

Policy information about <u>studies involving human research participants and Sex and Gender in Research.</u>

| | |
|---|---|
| Reporting on sex and gender | This large prospective cohort of 406,579 adults included 224,681 women and 181,898 men, with a mean [SD] age of 66.1 [8.4] years. All participants provided informed written consent to take part in the study and be followed-up through linkage to health-related records. The main analyses were assessed in subgroups based on sex and other population characteristics such as age, ethnicity, and BMI. |
| Population characteristics | In this prospective cohort of 406,579 adults (224,681 women, 181,898 men; mean [SD] age 66.1 [8.4] years), 26,181 had a positive test for SARS-CoV-2. The primary comparison cohorts comprised 26,181 participants in the SARS-COV-2 infection group, 380,398 in the contemporary control group and 384,030 in the historical control group. Detailed demographic and medical characteristics of all comparison cohorts before and after weighting were shown. For example, before weighting, participants in the infection group were younger (mean age: 66.0 years vs 68.8 years), less likely from the White ethnic group (84.6% vs 93.7%), more socioeconomically deprived (mean IMD: 20.5 vs 17.3), and more physically obese (mean BMI: 28.1 vs 27.3), compared with contemporary controls. After weighting, all covariates are balanced (ASMD<0.1). |
| Recruitment | The UK Biobank is an ongoing community-based prospective cohort study, which recruited more than 500,000 participants out of 9.2 million adults aged 40-69 years in the UK who were identified from National Health Service and invited to participants (5.5% response rate). The baseline survey took place from 2006 to 2010 in 22 assessment centers. |
| Ethics oversight | This study was based on data from UK Biobank. All participants provided written informed consent at the UK Biobank cohort recruitment. This study received ethical approval from UK Biobank Ethics Advisory Committee (EAC) and was performed under the application of 65397. |

Note that full information on the approval of the study protocol must also be provided in the manuscript.

# Field-specific reporting

Please select the one below that is the best fit for your research. If you are not sure, read the appropriate sections before making your selection.

☒ Life sciences ☐ Behavioural & social sciences ☐ Ecological, evolutionary & environmental sciences

For a reference copy of the document with all sections, see nature.com/documents/nr-reporting-summary-flat.pdf

# Life sciences study design

All studies must disclose on these points even when the disclosure is negative.

| | |
|---|---|
| Sample size | We included UK Biobank participants from England who were still alive by March 1, 2020 (N=406,579) to construct infected and control cohorts. We further excluded those with the history of mental health outcomes one year before the start of follow-up (327,022 participants in the contemporary control, 21,307 in the COVID-19 group, and 332,740 in the historical control). To our knowledge, the UK Biobank including about half a million participants is one of the current largest random well-controlled population-based cohort with detailed and robust recording of confounding factors that were largely unavailable in previous studies based on electronic health records. The large sample size obtained were deemed to provide reliable risk estimates of mental health outcomes. |
| Data exclusions | Participants with the history of mental health outcomes in one or two year before the start of follow-up were excluded to avoid potential reverse causality. |
| Replication | The main aim of the current study is to assess the risk of mental health outcomes after COVID-19 compared to uninfected control. In the main analyses, we used individuals with no evidence of SARS-CoV-2 infection but exposed similar pandemic-related environmental stressors as contemporary control group. For replication of major analyses, in the same study, we constructed a historical control cohort predating the pandemic and then compared the risk of mental health outcomes following SARS-CoV-2 infection compared with those unaffected by the COVID-19 pandemic. The results using the historical control cohort were consistent with the main analyses of contemporary control. |
| Randomization | No randomization was required as all samples were included in the analysis |
| Blinding | No blinding was applicable to this observational study as no intervention were applied to participants. |

# Reporting for specific materials, systems and methods

We require information from authors about some types of materials, experimental systems and methods used in many studies. Here, indicate whether each material, system or method listed is relevant to your study. If you are not sure if a list item applies to your research, read the appropriate section before selecting a response.

## Materials & experimental systems

| n/a | Involved in the study |
|-----|----------------------|
| ☒ | ☐ Antibodies |
| ☒ | ☐ Eukaryotic cell lines |
| ☒ | ☐ Palaeontology and archaeology |
| ☒ | ☐ Animals and other organisms |
| ☒ | ☐ Clinical data |
| ☒ | ☐ Dual use research of concern |

## Methods

| n/a | Involved in the study |
|-----|----------------------|
| ☒ | ☐ ChIP-seq |
| ☒ | ☐ Flow cytometry |
| ☒ | ☐ MRI-based neuroimaging |

