## [Peer Review File · Nature Human Behaviour]

Peer Review Information

Journal: Nature Human Behaviour

Manuscript Title: Long-term risk of psychiatric disorder and psychotropic prescription after SARS-CoV-2 infection among UK general population

Corresponding author name(s): Junqing Xie

Reviewer Comments & Decisions:

Decision Letter, initial version:

23rd January 2023

Dear Mr Xie,

Thank you once again for your manuscript, entitled "Long-term mental health outcomes after SARS-CoV-2 infection: prospective cohort study," and for your patience during the peer review process.

Your manuscript has now been evaluated by 3 reviewers, whose comments are included at the end of this letter. In the light of their advice, I regret that we cannot offer to publish your manuscript in Nature Human Behaviour.

While the reviewers find your work of some interest, they raise concerns about the appropriateness of the technical approach, the conceptual advance your findings represent over earlier work, and the strength of the novel conclusions that can be drawn at this stage. We feel that these reservations are sufficiently important as to preclude publication of this work in Nature Human Behaviour.

I am sorry that we cannot be more positive on this occasion but hope that you will find our reviewers' comments helpful when preparing your paper for submission elsewhere.

Sincerely,

Charlotte Payne

Charlotte Payne, PhD
Senior Editor
Nature Human Behaviour

Reviewer expertise:

Reviewer #1: neurologic outcomes post-COVID-19 in the US

Reviewer #2: cohort studies, COVID-19 long term outcomes

Reviewer #3: psychiatric outcomes post COVID-19 infection

Reviewers' Comments:

Reviewer #1:

Remarks to the Author:

This is a report that reproduces known findings on the relationship between covid and risks of mental health outcomes.

Here are few comments for your consideration.

The analysis on vaccinated and unvaccinated is deeply flawed. If the authors wanted to examine the effect of vaccination, a study optimized to address this question should be designed including making sure that infection among vaccinated (breakthrough infection) and among unvaccinated are aligned in space and time (because as the authors know, variants, treatments, etc.. change dramatically over space and time) and , for example, the comparison of unvaccinated covid cases in March 2020 to covid cases recently is a flawed comparison. Care setting also must be accounted for; since we know that vaccines reduce risk of severe disease/hospitalization/death. Please either do this analysis correctly or delete it.

The analyses comparing to other respiratory tract infections are not useful. A clear head-to-head comparison between covid and flu or other well-defined upper respiratory tract infection that are treated in the same care setting (comparison of outpatient to outpatient or hospitalized to hospitalized) is needed. The current "other respiratory tract infections" is a mixed bag of a lot of stuff. This is not good science. The current analyses are deeply flawed, no wonder they led to almost uninterpretable conclusion. This muddies the water and creates a lot of confusion. If better analyses cannot be done, it would be better to delete/remove flawed analyses than publish them.

I don't understand the motivating rationale to do the test negative design; it seems that the authors are acutely aware that testing for covid-19 is not random, and that test negative design studies are subject to confounding by testing indication (for example, cancer patients needing chemotherapy in June 2020 would be tested before they receive chemotherapy; most would test negative; if you examine their data vs positive people, you may find that negative test is associated with mortality; it does not mean that negative covid test causes mortality, it just means that there is huge confounding here for testing). In a way, the results confirm that the authors – despite their best effort – seem to have been unable to remove confounding. To me, this lessens my confidence in the results significantly. After reading this section, I took a closer look at the HR for all the other analyses in this manuscript and compared them to other published studies; I found the numbers to be too high – suggesting that the approach (while it imitates well designed studies, is likely riddled with a lot of confounding).

Throughout the narrative, the authors do not do justice to analyzing prior work and identifying a knowledge gap. They repeat themselves multiple times saying that prior work did not account for socioeconomic and lifestyle factors but many of the cited studies did. They also state that "the impact of vaccination status was not investigated in previous studies, and the association between vaccination and psychiatric sequelae of COVID-19 remain uncertain"; this statement (along with many others in the intro and discussion) is also factually incorrect.

I think the authors may want to consider framing the research inquiry to reproduce prior reports in the UK biobank.

Reviewer #2:

Remarks to the Author:

Due to an unforeseen work load during these busy winter months I will not be able to give this manuscript a

proper/full review. By giving it a brief look, I do think the authors have performed a very solid study complementing the current body of evidence very nicely! I do hope the authors thoroughly explain what advantages the UK Biobank data sources have in relation to national registries and EHR systems. Also, they could emphasize more how the potential selection bias of the UK Biobank participants might mean with regards to observed effect sizes etc. Furthermore, I also hope they give a proper description of the numbers of individuals with a previous history of each investigated outcome for the incident analyses.

Reviewer #3:

Remarks to the Author:

The authors present a retrospective cohort study using UK Biobank data to compare the incidence of psychiatric morbidity and prescriptions for psychotropic medications among individuals infected with COVID-19. This represents a valuable contribution to the literature. A few comments which may assist the authors in strengthening their manuscript.

-In the methods, I would point out that the cluster of ICD codes that the authors' identify as anxiety disorders also includes trauma-related disorders and somatoform disorders as these are distinguished from anxiety disorders in the DSM and this may confuse readers. The authors do address this later in the manuscript, but might consider a brief parenthetical statement about this in the methods.

-Prescription data was included for antidepressants, antipsychotics, benzodiazepines and opioids. I'm curious about the inclusion of opioids which are not typically used to treat any primary psychiatric disorder. The authors might address this point. I am also confused about the exclusion of lithium which is used almost exclusively to treat psychiatric illness (whereas many antidepressants are prescribed for a range of non-psychiatric indications) and thus would represent a good candidate for inclusion in the analysis. Furthermore, my assumption is that other mood stabilizers (e.g., anti-epileptics like valproic acid) were not included because of their use for non-psychiatric indications, but this should be addressed. In particular, their omission is confusing in light of the inclusion of opioids.

-If I understand correctly, the authors are denoting that psychiatric diagnoses they identify are new by the lack of corresponding prescriptions or diagnostic codes 1 year prior to the start of the study period. I would note in the discussion that the natural history of certain disorders might suggest that at least some of the cohort is having recurrent rather than de novo symptoms despite having not been treated/diagnosed in the prior year (e.g., MDD episodes can certainly occur with greater than a year in between).

-I am not qualified to comment on the nuances of the quantitative methodology but would recommend a biostatistician review and comment on these given the heavily quantitative nature of this manuscript.

-I am struggling to understand the result shared on line 322. I am inferring that the authors are comparing the COVID-19 infection group to the non-infection cohort subgroup with negative COVID-19 tests. I am confused by the language "were not significant or decreased." Please clarify.

-The authors' discussion paragraph about the finding that any respiratory infection was more strongly associated with new psychiatric diagnoses than was COVID-19 could be clearer. First of all, the authors posit that the individuals in the non-COVID respiratory infection cohort may have been sicker than the COVID-19 cohort. However, it was not clear to me why that would be the case; were the members of the non-COVID infection cohort all hospitalized? Or could they, too, have been diagnosed in the community setting? Furthermore, this discussion is combined in the same paragraph with a discussion about vaccination status and this makes it even more difficult to follow. I would encourage a reconfiguration of that element of the discussion to clarify the finding and articulate a possible explanatory hypothesis.

Minor Points:

-Line 90 should read: "All participants provided written informed consent at recruitment." You might also add "as described elsewhere" and provide a citation with further details about recruitment for the UK Biobank study.

-Line 90-91: Consider citing STROBE guidelines

-Line 96: Participants should be participate

-Throughout the document the authors refer to the infected cohort as infection group. I believe it would read better with a "the" before infection group. I would also encourage the authors to consider more descriptive language such as "the COVID-19 infected cohort"

-Line 377: I think that the authors might mean specifically rather than "specially"

-Line 378: WarrantS further research

I'm looking forward to seeing this manuscript in print and believe that my comments can all be addressed through discussion and will not require any changes to the experimental design or analysis. Thank you for this important contribution. With best regards,
Dan Shalev

Following suitable revisions, you may want to consider transferring your manuscript. I suggest that you consider Scientific Reports as a suitable venue for your work. To transfer your manuscript there, please use our manuscript transfer portal [LINK REDACTED**]. You will not have to re-supply manuscript metadata and files, unless you wish to make modifications, but please note that this link can only be used once and remains active until used. For more information, please see our manuscript transfer FAQ page.

Note that any decision to opt in to In Review at the original journal is not sent to the receiving journal on transfer. You can opt in to In Review at receiving journals that support this service by choosing to modify your manuscript on transfer. In Review is available for primary research manuscript types only.

Author Rebuttal to Initial comments

In support of our request, we are providing a point-by-point list what we believe to be the flawed aspects of the review in question with justifications for why we believe they are problematic and/or biased. **Note Reviewer 1's comments are in red type.**

1. **"This is a report that reproduces known findings on the relationship between covid and risks of mental health outcomes."**

The reviewer opens by saying the manuscript "reproduces known findings" immediately dismissing it as derivative and not particularly informative, important or worthy of publication.

The first two papers investigating the association between COVID-19 and subsequent psychiatric disorders both authored by Taquet, et al, were published in Lancet Psychiatry November, 2020 ^[2] and April, 2021. ^[3] By the reviewer's logic, any study coming after these two initial reports is simply a replication, not taking into account different data sources, population sizes, follow-up durations and many other factors that can make studies addressing the same question much more informative than simple replications. By this logic the subsequent publication by Yan Xie & Ziyad Al-Aly et al, even though with larger sample size and longer follow-up,^[4] is still simply **"a report that reproduces known findings"**, especially given that all three of these papers were based on electronic health records (EHR) or registry data from the USA. The "by setting" and "Comparison the psychiatric sequela of COVID-19 with other diseases like influenza" analyses by Yan Xie & Ziyad Al-Aly et al are both **not novel** and have been done in the two early Lancet Psychiatry papers. Therefore, an arbitrary labeling the manuscript as **"a report that reproduces known findings"** and ignoring its added value (which we will discuss in detail below) to the literature is unfounded, dismissive and

discriminatory. Our study investigated the long-term psychiatric sequela of COVID-19 and identified the beneficial effect of vaccination in the current largest reported population-based cohort of ~400,000 participants with a one-year follow-up.

2. “Throughout the narrative, the authors do not do justice to analyzing prior work and identifying a knowledge gap. They repeat themselves multiple times saying that prior work did not account for socioeconomic and lifestyle factors but many of the cited studies did.”

This criticism is factually incorrect and demonstrates a negative bias. In our submitted manuscript we clearly state “socioeconomic and lifestyle factors and public health interventions in the context of pandemic such as vaccination and quarantine measures that were associated with both SARS-CoV-2 infection and mental health conditions **were only crudely measured or not available** in these analyses”. We specifically did not state that prior studies failed to account for these factors at all.

In addition, for example, in the EHR study by the Yan Xie & Ziyad Al-Aly et al, the only lifestyle factor adjusted for was smoking status. Other important lifestyle factors such as drinking status and physical activity that are associated with the risk of both mental disorders and COVID-19 are not considered in previous HER study such as the one by the Yan Xie & Ziyad Al-Aly et al, but are available in the UKB and were adjusted to reduce confounding and potential bias.

In fact, accurate and clear conclusions on long-COVID cannot be derived from any small number of studies. Several additional analyses using the EHR data in the UK with similar PS-weighting/matching methods applied were published later after the initial two reports from the Taquet group mentioned above.

As for addressing knowledge gaps, we stated in the introduction and discussion, all studies using EHR data have an inherent bias^[5] and cannot represent the general population (participants must be enrolled in the system and receive care within the structure the EHR represents and findings are therefore not generalizable beyond that EHR population. Further, detailed measures of potential confounders were not often available within many EHR systems, leading to residual bias. Studies of different designs (like observational studies and randomized clinical trials) provide different levels of evidence. Well-designed and controlled randomized population-based cohort studies with detailed and robust recording of exposure and outcome and less bias and confounding can inform and potentially strengthen the findings initially reported in the earlier EHR studies.

3. “They also state that “the impact of vaccination status was not investigated in previous studies, and the association between vaccination and psychiatric sequelae of COVID-19 remain uncertain”; this statement (along with many others in the intro and discussion) is also factually incorrect.”

This criticism by the reviewer is factually incorrect and if further biased and dismissive of our long-term findings and their novel contribution to the literature. The association of vaccination with psychiatric sequela was to the best of our knowledge not evaluated in any of the cited literature available to us during the drafting of our manuscript. After the completion of our manuscript, two papers on the topic were published during May and July, 2022.^[4, 6]

However, each of these two studies employed only a 6-month follow-up, leaving the impact of vaccination at long-term follow-up, such as one-year, an unanswered question.

It was in this context of long-term follow-up that we stated in our introduction “questions remain on the incidence and absolute risk of psychiatric outcomes, and on the potential protective effect of vaccination.” We then assessed the effect of vaccination up to 1-year of follow-up and found the psychiatric risk after breakthrough infection were significantly reduced in a valid and large cohort representing the general population. This is an important finding that informs service planning during the post-pandemic era and a considerable advance on the previous literature.

4. “The analysis on vaccinated and unvaccinated is deeply flawed. If the authors wanted to examine the effect of vaccination, a study optimized to address this question should be designed including making sure that infection among vaccinated (breakthrough infection) and among unvaccinated are aligned in space and time (because as the authors know, variants, treatments, etc. change dramatically over space and time) and , for example, the comparison of unvaccinated covid cases in March 2020 to covid cases recently is a flawed comparison.”

This comment is factually incorrect. We never directly compared breakthrough infection (BTI) with non-breakthrough infection! In our analyses, we did compare the risk mental health outcomes of BTI or non-BTI with non-infection cohort separately and for both comparisons the start of follow-up was aligned as clearly described in our methods. A comparable approach has been reported previously. ^[6]

5. “The current “other respiratory tract infections” is a mixed bag of a lot of stuff. This is not good science.”

This statement is biased. Out of the 4 good quality papers cited in the introduction to profile previous research progress ^[2-4,7] three papers specifically used the term and reported on “other respiratory tract infections”. We used the same ICD-10 codes for RTIs as these studies both to report results and to facilitate comparison with these previous studies. We consider this approach “good science”!

6. “The current analyses are deeply flawed, no wonder they led to almost uninterpretable conclusion.”

We are hard pressed to respond to this broad, sweeping unsubstantiated, and pejorative “criticism.” This comment once again shows an explicit bias on the part of the reviewer against our work shows bias, and approaches an ad hominem attack. These biased comments are offered without supporting examples and appear as total conjecture by the reviewer. The definition of whether research or analyses is good science should be based on the concrete evidence. It’s irresponsible, and frankly insulting to claim our results are “not good science” without a clear explanation of why they are not.

Based on the published literature, our results as discussed in the manuscript are consistent with two UK EHR studies suggesting the increase in risk was larger for individuals with non-COVID-19 respiratory tract infections than individuals with SARS-CoV-2 infection.^[7,8] In addition, Taquet et al who initially reported increased risk of psychiatric disorders after COVID-19 compared with influenza or RTIs ^[3] found the risks of psychiatric disorders after COVID-19 were lower than influenza or RTIs from 2 months to the 2-year follow-up in their updated publication,^[9] consistent with our reported

findings. Ours are clearly not uninterpretable findings drawn from deeply flawed analyses but rather in alignment with the findings of previous publications. We cannot help but hear this reviewer grinding their own personal axe!

7. "I don't understand the motivating rationale to do the test negative design; it seems that the authors are acutely aware that testing for covid-19 is not random, and that test negative design studies are subject to confounding by testing indication (for example, cancer patients needing chemotherapy in June 2020 would be tested before they receive chemotherapy; most would test negative; if you examine their data vs positive people, you may find that negative test is associated with mortality; it does not mean that negative covid test causes mortality, it just means that there is huge confounding here for testing). In a way, the results confirm that the authors – despite their best effort – seem to have been unable to remove confounding. To me, this lessens my confidence in the results significantly.

The rationale for the test negative design was to provide caution about the causal effect of COVID-19 infection on mental health to help reader and broader public avoid interpreting the findings as causality. We note that this did not both either of the other two reviewers.

There is good evidence for strong confounding in analyses of those who present for COVID-19 testing; and this is likely to bias studies of COVID-19 and psychiatric sequelae.^[10] Consistent with our finding that no increased risk of mental health outcomes were observed in test-positive vs test-negative participants, a large UK national EHR study^[8] (test-positive participants vs those without positive results: adjusted HR 1.83, 1.66-2.02; test-negative participants vs those without positive results: adjusted HR 1.71, 1.66-1.77; the similar level of HR suggesting the risk among persons with test-positive vs test-negative is alike despite not direct comparisons performed by this study) and a study of Danish registry data^[10] (aRR depression 0.91, 0.46 to 1.80; aRR anxiety disorders 0.54, 0.32 to 0.90) both found no evidence of elevated risk of subsequent psychiatric outcomes among those with a positive COVID-19 test. However, considering the nature of registry or EHR data used, these biases are not quantified but are highly likely in the four published US EHR studies including the one by reviewer and one EHR study in the UK.^[2-4,7,9] Thus, we recommended that such findings when reported to the public should be interpreted with caution and the need for further research be made clear. Similarly, we noted that the use of random population-based cohorts that might be less subject to confounding bias is also recommended to be used to improve the understanding of whether COVID-19 infection directly increases the risk of mental illness.^[9]

Therefore, it is inappropriate for the reviewer to claim that "seem to have been unable to remove confounding" Actually, no publications on this topic have successfully removed all potential confounding, such as the one caused by test-negative participants and residual confounding as a result of lack of comprehensive lifestyle and other factors. For example, the paper by Yan Xie & Ziyad Al-Aly et al and several other US EHR studies also included a considerable number of test-negative participants but did not conduct the similar analyses, bias may also exist in these EHR data (including the one by Yan Xie & Ziyad Al-Aly et al) and influence their results. The unadjusted covariates such as several important demographic and lifestyle factors (as mentioned in the response 2) may lead to residual confounding.

In our study, we have made a good faith effort but even then described potential unremoved confounding by test-negative participants as a clear limitation of the study. In addition, findings in our study and some other studies indicating that these risks of mental health outcomes were higher in those with more severe disease and were significantly reduced in fully vaccinated people strengthen the causal inference. However, we believe the potential confounding caused by the test behavior may still exist in ours and all study of head-to-head comparison between COVID-19 and test-negative/no

evidence of COVID-19/other diseases and this concern should be clearly stated in the discussion to inform readers and policy maker.

8. "...analyses in this manuscript and compared them to other published studies; I found the numbers to be too high – suggesting that the approach (while it imitates well designed studies, is likely riddled with a lot of confounding)."

It is unclear to us exactly what "numbers" did the reviewer find "to be too high." A bit of specificity here would have allowed us to provide a more cogent response. However, we will respond with the assumption that the reviewer meant the risk estimates such as HR.

Direct comparison (especially with the aim of identifying "confounding") between studies with different study designs, baseline characteristics (e.g. mean age and the prevalence of chronic diseases etc.), diagnostic criteria (e.g. between US and UK), property of participants (some studies included cases tested in community and hospital but some were restricted to the severe cases tested diagnosed in the hospital setting), outcome assessed (composite mental health outcomes or subtypes such as depression and anxiety), variable adjusted etc. is deeply flawed. As the basic knowledge of epidemiology is the balance of variables or characteristics between the groups of comparison. These obvious factual errors by the reviewer also significantly decreased my confidence on their expertise and righteous attitude toward peer review-which should have been a fair and evidence-based process aiming to promote the advance of natural science. In addition, following reviewer's logic, the range of HRs (Test positive participants vs those without evidence of infection; composite mental health outcome) published in previous literature is from about 1.4 to 2.4, and our HR is 2.0. Again, the claim of "to be too high" tends to be non-evidence based and factually incorrect and are the review's subjective/biased judgement.

9. "...the approach (while it imitates well designed studies, is likely riddled with a lot of confounding)."

The reviewer makes a global, depreciating and unsubstantiated claim. Our study simultaneously is well designed, likely because it merely "imitates well designed studies" rather than "employs designs comparable to" or other less pejorative turns of phrase. But even a well designed study achieved through mundane "imitation" has only resulted in a study "likely riddled with a lot of confounding." All of this from the reviewer without a word of substantiation. We are left totally confused as to how a study might be well designed, even if merely through "monkey see – monkey do" imitation and yet be "riddled with a lot of confounding." Our minds boggle at this reviewer's repeated, unsubstantiated depreciating tone.

In conclusion, we believe we have demonstrated that Reviewer 1's evaluation was a mix of factually incorrect criticisms, many of them being broad, pejorative, and unsubstantiated claims. Where the reviewer has provided sufficient clarity of concern we have provided cogent responses, which we believe demonstrate the soundness of our study design, data analyses and conclusions. Given this we again ask that you reconsider our submission after obtaining an additional more objective and less biased review.

Thank you for your attention to this matter. We look forward to your reply.

Sincerely,

All Authors

Reference

1. How (not) to appeal. *Nat Hum Behav* 5, 805–806 (2021). <https://doi.org/10.1038/s41562-021-01174-w>
2. Taquet M, Luciano S, Geddes JR, Harrison PJ. Bidirectional associations between COVID-19 and psychiatric disorder: retrospective cohort studies of 62 354 COVID-19 cases in the USA [published correction appears in *Lancet Psychiatry*. 2021 Jan;8(1):e1]. *Lancet Psychiatry*. 2021;8(2):130-140.
3. Taquet M, Geddes JR, Husain M, Luciano S, Harrison PJ. 6-month neurological and psychiatric outcomes in 236 379 survivors of COVID-19: a retrospective cohort study using electronic health records. *Lancet Psychiatry*. 2021;8(5):416-427. doi:10.1016/S2215-0366(21)00084-5
4. Xie Y, Xu E, Al-Aly Z. Risks of mental health outcomes in people with covid-19: cohort study *BMJ* 2022; 376 :e068993 doi:10.1136/bmj-2021-068993
5. Agniel D, Kohane I S, Weber G M. Biases in electronic health record data due to processes within the healthcare system: retrospective observational study *BMJ* 2018; 361 :k1479 doi:10.1136/bmj.k1479
6. Al-Aly, Z., Bowe, B. & Xie, Y. Long COVID after breakthrough SARS-CoV-2 infection. *Nat Med* 28, 1461–1467 (2022). <https://doi.org/10.1038/s41591-022-01840-0>
7. Clift AK, Ranger TA, Patone M, et al. Neuropsychiatric Ramifications of Severe COVID-19 and Other Severe Acute Respiratory Infections. *JAMA Psychiatry*. 2022;79(7):690–698. doi:10.1001/jamapsychiatry.2022.1067
8. Abel KM, Carr MJ, Ashcroft DM, et al. Association of SARS-CoV-2 Infection With Psychological Distress, Psychotropic Prescribing, Fatigue, and Sleep Problems Among UK Primary Care Patients. *JAMA network open*. 2021;4(11):e2134803.
9. Taquet M, Sillett R, Zhu L, et al. Neurological and psychiatric risk trajectories after SARS-CoV-2 infection: an analysis of 2-year retrospective cohort studies including 1 284 437 patients. *Lancet Psychiatry*. 2022;9(10):815-827. doi:10.1016/S2215-0366(22)00260-7
10. Lund LC, Hallas J, Nielsen H, et al. Post-acute effects of SARS-CoV-2 infection in individuals not requiring hospital admission: a Danish population-based cohort study. *The Lancet Infectious diseases*. 2021;21(10):1373–1382.
11. <https://www.bmj.com/content/376/bmj-2021-068993/rr-1>; Matthias R Pierce. Caution about causal effect of COVID-19 infection on mental health.

Decision Letter, first revision:

Dear Mr Xie,

Thank you for your correspondence asking us to reconsider our decision on your Article, "Long-term mental health outcomes after SARS-CoV-2 infection: prospective cohort study". I apologise again for the delay in my reply. After careful consideration, we have decided that we would be willing to consider a revised version of your manuscript.

Along with your revised manuscript, you should also submit a separate point-by-point response to all of the concerns

raised by the referees, in each case describing what changes have been made to the manuscript or, alternatively, if no action has been taken, providing a compelling argument for why that is the case. Please ensure your responses to all comments, including those of Reviewer 1, are constructive and professional. Please provide explanations, justifications and clarifications as warranted, and ensure these are also reflected in changes to the manuscript file. This is essential for all reviewers to fully understand the history of the work, and to evaluate your responses to the previous round of review.

If we feel that a substantial attempt has been made to address the referees' comments, this response will be sent back to the referees - along with the revised manuscript - so that they can judge whether their concerns have been addressed satisfactorily or otherwise.

I should stress, however, that we would be reluctant to trouble our referees again unless we thought that their comments had been addressed in full.

- ensure it complies with our format requirements as set out in our Guide to Authors.
- state in a cover note the length of the text, methods and figure legends; the number of references and the number of display items.

Please ensure that all correspondence is marked with your Nature Human Behaviour reference number in the subject line.

Please use the following link to submit your revised manuscript:

[REDACTED]

We hope to receive your revised paper within four weeks. If you cannot send it within this time, please let us know so that we can close your file. In this event, we will still be happy to reconsider your paper at a later date so long as nothing similar has been accepted for publication at Nature Human Behaviour or published elsewhere in the meantime. Should you miss the four-week deadline and your paper is eventually published, the received date will be that of the revised, not the original, version.

I look forward to hearing from you soon.

Best regards,

Charlotte Payne

Charlotte Payne, PhD
Senior Editor
Nature Human Behaviour

Author Rebuttal, first revision:

Response to reviewers' comments:**Critical Updates:**

1. With the latest release of the UK Biobank health record linkage data, we are now able to follow all participants for one year by extending the study ending date from 2021/09/30 to 2022/09/30 in the revised version. For example, those diagnosed with COVID-19 on 2021/03 originally censored at 2021/09 are now censored at 2022/03. As a result, the median follow-up duration in both infected and control cohorts have increased by more than 100 days from 250 to 365 days, allowing us to better assess long-term psychiatric sequelae. Our analyses of breakthrough/non-breakthrough infection during the 1-year follow-up have significant public health implications, as these associations have only been assessed during a maximum of 6-month of follow-up in previous EHR-based studies. These changes are now reported throughout the manuscript.
2. We have updated the recent evidence on COVID-19 psychiatric sequelae published after the completion of our initial manuscript. High-quality studies on this topic have been added, summarized, and compared in the discussion, including in particular two studies that specifically assessed the impact of vaccination on COVID-19 psychiatric sequelae, albeit both with maximal 6-month follow-ups.
3. We have used a more stringent two-year washout period when studying incident outcomes throughout the manuscript and included "mood stabilizers" as a category of psychotropic prescriptions as suggested by the third reviewer.

Reviewer #1:

Remarks to the Author:

This is a report that reproduces known findings on the relationship between covid and risks of mental health outcomes.

Response:

The first two papers investigating the association between COVID-19 and subsequent psychiatric disorders both authored by Taquet, et al, were published in Lancet Psychiatry November, 2020¹ and April, 2021.² It is unfair to claim that any study that comes after a previous one is a simple replication without taking into account possible different data sources, different health care pattern between countries, population sizes, follow-up periods, and many other factors of interest. We have seen the subsequent studies published by Yan Xie & Ziyad Al-Aly et al.³ using similar methods and data sources (but with larger sample size and longer follow-up), and argue that an arbitrary labeling a study as "a report that reproduces known findings" and ignoring its added value (which we will discuss in detail below) to the existing evidence is unfounded and dismissive. Our study reports the currently largest prospective cohort based on the general population, rather than registry data with their inherent limitations, our study comprehensively characterizes the long-term mental sequelae after COVID-19 and the potential protective effect of full vaccination on them. Notably, our study demonstrates that the breakthrough infection had a minimal impact on long-term psychiatric sequelae by following participants up to 1 year (the longest follow-up period to our best knowledge in literature), which is of particular novelty and will have important public health implications.

Here are few comments for your consideration.

The analysis on vaccinated and unvaccinated is deeply flawed. If the authors wanted to examine the effect of vaccination, a study optimized to address this question should be designed including making sure that infection among vaccinated (breakthrough infection) and among unvaccinated are aligned in space and time (because as the authors know, variants, treatments, etc.. change dramatically over space and time) and, for example, the comparison of unvaccinated covid cases in March 2020 to covid cases recently is a flawed comparison. Care setting also must be accounted for; since we know that vaccines reduce risk of severe disease/hospitalization/death. Please either do this analysis correctly or delete it.

Response:

This comment is factually incorrect as we did not directly compare breakthrough infection (BTI) with non-BTI in our study. In the main analyses, we compared the risk of the BTI and non-BTI with non-infection cohort separately, intentionally to avoid the issues raised by the reviewer, and the start of follow-up was aligned for both comparisons as described in our methods and results. A comparable approach has been used by Xie & Ziyad Al-Aly et al previously.⁴ In the infection group, only a small fraction of participants tested positive in the hospital setting (~2500), the vaccination analyses further stratified by test setting would lack statistical power. However, we estimated the risk stratified by care settings (community-based infection case vs hospitalized cases) and reported each HR. Please see Figure 6. We also simultaneously adjusted for health care utilization such as hospital admissions and a set of clinically informed or data-driven covariates to account for potential confounding.

Our results on vaccination have significant public health implication. Only two large electronic healthcare records (EHR) studies reported a reduced risk of psychiatric sequelae after vaccination during a maximum of six months' follow-up,^{4,5} the effect of vaccination on the full spectrum of mental health outcomes over longer follow-up periods, such as one year, remains unclear. Our findings further demonstrated that full vaccination before infection may reduce the psychiatric sequelae of COVID-19 as proxied by psychiatric diagnosis or psychotropic prescription during a much longer 1 year follow-up. In addition, we found that clinically diagnosed psychiatric disorders after breakthrough infection was similar to that of vaccinated participants with no evidence of infection during the extended one-year follow-up period. However, the slight increase in psychotropic prescription in primary care, where non-specific symptoms of mental health problems such as psychological distress (e.g., symptoms of stress, depression, and anxiety) are likely to be the main driver, indirectly suggests that the excess burden of subclinical mental health problems related to breakthrough infection may still exist at one year. We now discuss and compare our results directly with previous studies of vaccination and mental health or long-COVID symptoms in the manuscript.

The analyses comparing to other respiratory tract infections are not useful. A clear head-to-head comparison between

covid and flu or other well-defined upper respiratory tract infection that are treated in the same care setting (comparison of outpatient to outpatient or hospitalized to hospitalized) is needed. The current “other respiratory tract infections” is a mixed bag of a lot of stuff. This is not good science. The current analyses are deeply flawed, no wonder they led to almost uninterpretable conclusion. This muddies the water and creates a lot of confusion. If better analyses cannot be done, it would be better to delete/remove flawed analyses than publish them.

Response:

The definition of whether research or analyses is good science should be based on concrete evidence. Out of the 4 papers of good methodological quality cited in the introduction to profile previous research progress,^{1-3,6} three papers specifically used the term and reported on “other respiratory tract infections (RTIs)”.^{1,2,6} We used the same ICD-10 codes for RTIs as these studies both to report results and to facilitate comparison with these previous studies. We consider this approach “good science”. In the original manuscript, our findings were clearly not “uninterpretable findings drawn from deeply flawed analyses” but rather were in alignment with the findings of previous publications. Based on the published literature, our results as discussed in the manuscript are consistent with two UK EHR studies suggesting the increase in risk was larger for individuals with non-COVID-19 respiratory tract infections than individuals with SARS-CoV-2 infection. In addition, Taquet et al who initially reported increased risk of psychiatric disorders during the six months after COVID-19 compared with influenza or RTIs, found the risks of psychiatric disorders after COVID-19 were lower than influenza or RTIs from 2 months to the 2-year follow-up in their updated publication, consistent with our reported findings of long-term mental health outcomes in the original manuscript.

Nevertheless, we agree with the reviewer that an even clearer head-to-head comparison between hospital-based COVID-19 and hospital-based other respiratory infections might be more useful. However, as mentioned above, only a small fraction of participants tested positive in the hospital setting, limiting the statistical power of such analyses. In addition, only a small number of influenza cases can be identified in our cohort as a contemporary control. Given these factors, we decided to remove this analysis. This analysis was originally secondary and independent of the main analyses between the infection group and contemporary/historical controls. Therefore, the main results and conclusions remain unaffected.

I don't understand the motivating rationale to do the test negative design; it seems that the authors are acutely aware that testing for covid-19 is not random, and that test negative design studies are subject to confounding by testing indication (for example, cancer patients needing chemotherapy in June 2020 would be tested before they receive chemotherapy; most would test negative; if you examine their data vs positive people, you may find that negative test is associated with mortality; it does not mean that negative covid test causes mortality, it just means that there is huge confounding here for testing). In a way, the results confirm that the authors – despite their best effort – seem to have been unable to remove confounding. To me, this lessen my confidence in the results significantly. After reading this section, I took a closer look at the HR for all the other analyses in this manuscript and compared them to other published studies; I found the numbers to be too high – suggesting that the approach (while it imitates well designed studies, is likely riddled with a lot of confounding).

Response:**Section A: why we conduct the analyses of test-negative control?**

We agree that the test-negative population may not be an ideal control group, as reasoned by the reviewer, and we therefore demoted it to as an explorative analysis in our study. However, we insisted to transparently report this part of results and spent some space in the discussion in order to (1) to quantify the extent to which using the test-negative cohort may impact the risk estimate, and (2) raise cautions for future studies that intend to use this design. See below for a detailed, evidence-based explanation of items (1) and (2).

Regarding the reviewer's point that "for example, cancer patients needing chemotherapy in June 2020 would be tested before receiving chemotherapy", we think this statement is a bit arbitrary, as studies based on real-world data theoretically vary depending on the real-world settings where/when/how the data were collected. In these previous EHR-based studies,^{1-3,7} it is likely that the test-negative cohort includes a substantial proportion of cancer or other critical patients. However, this limitation may not equally true for our study based on a community-based population. In the UK, nationwide infection screening among the general population was common during our study period.

- (1) There is good evidence for strong confounding in analyses of those who present for COVID-19 testing; and this is likely to bias studies of COVID-19 and psychiatric sequelae. Consistent with our finding that no increased risk of mental health outcomes were observed in test-positive vs test-negative participants, a large UK national EHR study⁸ (test-positive participants vs those without positive results: adjusted HR 1.83, 1.66-2.02; test-negative participants vs those without positive results: adjusted HR 1.71, 1.66-1.77; the similar level of HR suggesting the risk among persons with test-positive vs test-negative is alike despite not direct comparisons performed by this study) and a study of Danish registry data⁹ (aRR depression 0.91, 0.46 to 1.80; aRR anxiety disorders 0.54, 0.32 to 0.90) both found no evidence of elevated risk of subsequent psychiatric outcomes among those with a positive COVID-19 test. Similarly, a recent nationwide EHR study from Israel¹⁰ suggested that compared with participants with a negative PCR test, the risk of depression and anxiety were not differential and the risk of psychosis was decreased in those with a positive test during the long-term follow up (180-360 days after infection).¹⁰
- (2) However, considering the nature of registry or EHR data used, these biases are not quantified but are highly likely in the published US EHR studies including the largest one by Xie & Ziyad Al-Aly et al.^{1-3,7} Thus, we recommended that such findings when reported to the public should be interpreted with caution and the need for further research be made clear. In addition, the use of random population-based cohorts that might be less subject to confounding bias is also recommended to be used to improve the understanding of whether COVID-19 infection directly increases the risk of mental illness (<https://www.bmj.com/content/376/bmj-2021-068993/rr-1>).

It is also inappropriate for the reviewer to claim that "seem to have been unable to remove confounding." Actually, no publications on this topic have successfully removed all potential confounding, such as the one caused by test-negative

participants and residual confounding as a result of lack of comprehensive lifestyle and other factors. For example, the paper by Yan Xie & Ziyad Al-Aly et al and several other US EHR studies also included a considerable number of test-negative participants but did not conduct the similar analyses, bias may also exist in these EHR data and influence their results. The unadjusted covariates such as several important demographic and lifestyle factors may lead to residual confounding.

It may not appropriate to claim “numbers to be too high.” First, direct comparison, especially with the aim of identifying “confounding”, between studies with different study designs (excluding 1 year or 2 year history of mental disorders at baseline), diagnostic criteria (e.g. between US and UK), nature of participants (community-based or hospital based), outcomes assessed (composite mental health outcomes or subtypes such as depression and anxiety), and variables adjusted for etc. is not valid. Second, the range of HRs (Test positive participants vs those without evidence of infection; For composite mental health outcome) published in previous literature is from about 1.4 to 2.4, and our HR is 1.5. Again, the claim of “to be too high” tends to be non-evidence based therefore inappropriate.

Section B: What further analysis do we conduct to reduce concerns about bias and increase confidence in the results?

To further reduce the reviewer’s concerns about bias, we use the settings of positive outcome control (to test whether our study reproduces established knowledge about long COVID; e.g., fatigue after COVID-19, one of the common long-COVID symptoms defined by WHO) and negative outcome control (e.g. risk of skin neoplasms after COVID-19 where no associations were expected according to previous knowledge). The successful use of negative outcome controls can reduce concerns about the presence of potential biases related to study design, analytical methods, residual confounding, and other latent sources of bias. This approach has been used in previous epidemiological studies.³

Specifically, we use:

Positive outcome controls (fatigue [G93.3 and R53] and dyspnea [R06.0])

Negative outcome controls (neoplasms of skin [C43-C44], follicular cysts of skin [L72.0-L72.1])

Results showed that, compared with the contemporary control group without evidence of infection, the infection group had an increased risk of fatigue (HR, 2.76, 95% CI 2.38–3.20) and dyspnea (HR, 2.98, 95% CI 2.59–3.44). No significant associations between COVID-19 and any of the negative outcome controls were observed (neoplasms of skin (HR, 0.86, 95% CI 0.69–1.06); Skin cyst (HR, 1.41, 95% CI 0.91–2.20)).

Overall, we have made good faith efforts to control potential biases and provide robust risk estimates, including the use of both predefined and data-driven covariates to adjust for the difference in baseline characteristics between comparison groups, the use of a large-scale, prospective, population-based cohort with long-term follow up, and the use of robust and accurate definitions of exposure and outcomes. Additionally, our results were robust in multiple

sensitivity analyses and survived the rigorous application of negative controls. As with all observational studies, it is impossible to eliminate all confounding, which we acknowledge in the limitations section, and we do not recommend making causal inferences from ours or any other observational studies of long COVID. Nevertheless, given the lack of robust Mendelian randomization studies due to inadequate instrumental variables associated with SARS-CoV-2 infection, observational studies based on large population cohorts provide an informative reference.

Importantly, we also note an important source of bias related to the testing behavior to inform readers and future studies. In our study and many other publications, the test-negative group had even higher risk of mental health outcomes compared with test-positive group, suggesting that testing behavior could at least partially account for the observed increase in mental health outcomes after COVID-19. Previous EHR studies did not assess the risk in the test-negative vs test-positive groups or exclude individuals with negative test results in the control group. However, we believe that the potential confounding caused by the test behavior may still exist in ours and all such studies of head-to-head comparison between COVID-19 and other control groups and this concern should be clearly stated in the discussion to inform readers and policy makers.

These responses are all incorporated into the revised discussion section.

Throughout the narrative, the authors do not do justice to analyzing prior work and identifying a knowledge gap. They repeat themselves multiple times saying that prior work did not account for socioeconomic and lifestyle factors but many of the cited studies did. They also state that “the impact of vaccination status was not investigated in previous studies, and the association between vaccination and psychiatric sequelae of COVID-19 remain uncertain”; this statement (along with many others in the intro and discussion) is also factually incorrect.

Response:

In our submitted manuscript we clearly state, “socioeconomic and lifestyle factors and public health interventions in the context of pandemic such as vaccination and quarantine measures that were associated with both SARS-CoV-2 infection and mental health conditions were only crudely measured or not available in these analyses”. We specifically did not state that prior studies failed to account for these factors at all.

For example, in the largest EHR-based study by the Yan Xie & Ziyad Al-Aly et al, the only lifestyle factor adjusted for was smoking status, as shown in their articles. Other important lifestyle factors such as drinking status and physical activity, which are associated with the risk of both mental disorders and COVID-19, were not considered in the study by Yan Xie & Ziyad Al-Aly et al, but are available in the UKB and were adjusted to reduce confounding and potential bias. Another example is that, in the early EHR-based studies by Taquet, et al, the only socioeconomic covariate adjusted is socioeconomic deprivation, with no other lifestyle factors adjusted.

As for addressing knowledge gaps, we stated in the introduction and discussion, all studies using EHR data have an

inherent bias^{11,12} and cannot represent the general population (participants must be enrolled in the system and receive care within the structure the EHR represents and findings are therefore not generalizable beyond that EHR population. Further, detailed measures of potential confounders were not often available within many EHR systems, leading to residual bias. Studies of distinctive designs (like observational studies vs randomized clinical trials) provide distinctive levels of evidence. Well-designed and controlled randomized population-based cohort studies with detailed and robust recording of exposure and outcome and less bias and confounding can inform and potentially strengthen the findings initially reported in the earlier EHR studies. In summary, triangulation of evidence from different studies is particularly important in the context of global COVID-19 research.

There are also knowledge gaps in the effect of vaccination on the full spectrum of mental health outcomes over longer follow-up periods, such as one year. Only two large EHR studies reported a reduced risk of psychiatric sequelae after vaccination during a maximum of six months' follow-up. In the revised manuscript, we now discuss and compare our results on breakthrough/non-breakthrough infection during one-year follow-up with previous EHR studies of vaccination and mental health or long-COVID symptoms using a shorter follow-up.

I think the authors may want to consider framing the research inquiry to reproduce prior reports in the UK biobank. We thanks the reviewer for suggestions to improve the manuscript.

Reviewer #2:

Remarks to the Author:

Due to an unforeseen work load during these busy winter months I will not be able to give this manuscript a proper/full review. By giving it a brief look, I do think the authors have performed a very solid study complementing the current body of evidence very nicely! I do hope the authors thoroughly explain what advantages the UK Biobank data sources have in relation to national registries and EHR systems. Also, they could emphasize more how the potential selection bias of the UK Biobank participants might mean with regards to observed effect sizes etc. Furthermore, I also hope they give a proper description of the numbers of individuals with a previous history of each investigated outcome for the incident analyses.

Response: Advantages the UK Biobank data sources have in relation to national registries and EHR systems.

We thank the reviewer for these suggestions.

In the Introduction and Discussion section we stated that all studies using EHR data have an inherent bias^{11,12} and cannot represent the general population (participants must be enrolled in the system and receive care within the

structure the EHR represents and findings are therefore not generalizable beyond that EHR population. Further, detailed measures of potential confounders were not often available within many EHR systems, leading to residual bias. Studies of distinctive designs (like observational studies and randomized clinical trials) provide distinctive levels of evidence. Well-designed and controlled randomized population-based cohort studies with detailed and robust recording of exposure and outcome and less bias and confounding can inform and potentially strengthen the findings initially reported in the earlier EHR studies. We have rephrased the sentence in the introduction and discussion as:

Introduction: “Electronic health record (EHR) or registry data that is less representative of the general population (recruitment is dependent on utilization of the health system) and more vulnerable to latent bias, confounding, and the potential impact of the disruptions in health care services during the pandemic compared with prospective cohort based on the general population.”

Discussion: “In addition, recruitment in the EHR study is dependent on patients’ utilization of the health system, which vary substantially across countries with different health care delivery systems and limits the generalizability of findings beyond the population covered by the respective EHR database. For example, a large US study of psychiatric sequelae based on EHR data from discharged veterans, 90% of whom were male, was unlikely to be representative to the general population. The estimates of absolute risk difference or disease burden is particularly susceptible to the selected samples due to the difference in their baseline risk.”

In addition, we now provide further information in the Supplement to better illustrate the potential methodological pitfalls in study of long-COVID using EHR or registries compared with prospective cohort based on general population.

Supplementary Table 9. Methodological pitfalls in long-COVID study using electronic health records compared with community-based cohort study

	EHR-based study (MVP,¹ TriNetX,² and QResearch³)	Community-based cohort study (UK Biobank⁴)	Limitations of EHR compared with population cohort on result interpretation[†]
Population selection	 - Participants must be enrolled in the system and receive care within the structure the EHR represents 	 - Participants were randomly recruited at baseline and more representative of the general population 	 · Prone to sample selection bias · Unspecific target population · Findings had limited generalizable beyond that EHR population
Study design	 - Retrospective (looking backward) - Routinely collected data was retrospectively reviewed - A relatively large sample size[‡] 	 - Prospective (looking forward) - Participants were prospectively followed up for study outcomes - A modest to large sample size 	 · Outcome have already occurred in some of participants before study design · Prone to recall bias or misclassification bias · Causal interpretation of the findings is challenging · Inferior level of evidence compared with prospective cohort studies
Definitions of variables	 - Variables were defined after study design and extracted from data that were previously recorded for reasons not relating to the project. Some needed factors may not available in the records - Variables may be difficult to measure correctly or have large inter-observer variability 	 - A series of variables were measured using standardized methods such as questionnaire, including detailed lifestyle, socioeconomic, medication use, and comorbidity factors 	 · Poor control over the exposure, covariates, and potential confounders · Not all outcomes and exposures are formally adjudicated, as the database is dependent on coding by individual practitioners · Inadequate adjustment of important confounders may result in residual confounding

Vulnerability to the effect of COVID-19 pandemic

- Due to the disruptions in patient care and reduced availability of services during the pandemic, the help-seeking behaviors of some individuals may be changed, such as avoiding hospital/outpatient care

- Participants were recruited and started to follow up before the pandemic

· Vulnerable to recording or detection bias in the setting of pandemic
 · The effect of pandemic on help-seeking behaviors or access to health care may also impact cohort study if the outcomes are identified through data linkage to health registry

[†]Although some of these biases may also affect traditional population-based cohort, they are more common in research using EHR data. The potential for confounding is arguably larger in EHR data because the quality is often lower than that of traditional cohort. Note that the actual magnitude of unmeasured confounding likely depends on the comparison that is being made (e.g., COVID-19 vs non-infection/test-negative group here). In addition, confounding only plays a role in studies that aim to investigate causal effects.

[‡]Large sample sizes generally do not affect underlying biases, although they do however increase the precision of statistical tests, meaning that biased results are more likely to become statistically significant.

¹**Million Veteran Program (MVP, US):** Xie, Y., et al. Risks of mental health outcomes in people with covid-19: cohort study. *BMJ*.

²**TriNetX (US):** Taquet, M., et al. 6-month neurological and psychiatric outcomes in 236 379 survivors of COVID-19: a retrospective cohort study using electronic health records. *Lancet Psychiatry*.

³**QResearch (UK):** Clift, A. K. et al. Neuropsychiatric Ramifications of Severe COVID-19 and Other Severe Acute Respiratory Infections. *JAMA Psychiatry*.

⁴**UK Biobank (UK):** Yunhe Wang, et al. Long-term mental health outcomes after SARS-CoV-2 infection: prospective cohort study.

Response: Potential selection bias of the UK Biobank participants might mean with regards to observed effect sizes

We thank the reviewer for these suggestions.

Concerns have been raised that participants in UK Biobank may be suboptimally representative of the entire population in the UK and were likely to be older and generally healthier. These issues primarily affect the estimates of absolute incidence rates. Despite the relative risk between comparison groups was largely not influenced, this might still limit the generalizability of our findings to other younger populations. We have acknowledged this in the limitations section and cited an additional reference [Batty G D, et al. Comparison of risk factor associations in UK Biobank against representative, general population based studies with conventional response rates: prospective cohort study and individual participant meta-analysis. *bmj*, 2020, 368].¹³

Response: Numbers of individuals with a previous history of each investigated outcome for the incident analyses

We thank the reviewer for these suggestions.

We now provide the number of individuals with a history of each mental health outcome in the past two years before the start of follow up excluded in the analyses of incident outcomes (the infection group vs contemporary control).

Mental health outcome	Number [†]
Psychiatric diagnoses[#]	
Psychotic disorders	484
Mood disorders	8254
Mania/Bipolar affective disorder	541
Depressive episode	7794
Anxiety disorders	7217
Phobic anxiety disorders	534
Panic disorder	514
Generalized anxiety disorder	6561
Posttraumatic stress disorder	177
Substance use disorders	6558
Alcohol use disorder	1795
Tobacco use disorder	5109
Sleep disorders	3225
Psychotropic medications[#]	
Antipsychotics	290
Antidepressant	25662
SSRI	22781
SNRI	3265
Other antidepressant drugs	9243
Benzodiazepines	13628
Mood stabilizers	16998
Opioids	32127
Composite outcomes[#]	
Any psychiatric diagnosis	20185

Any prescriptions for psychotropic medications	76351
Any mental health related conditions	85687

[#]There is overlap in numbers as participants may have history of multiple psychiatric diagnoses or psychotropic medications.

[†]Participants were excluded from the incident analyses (the infection group vs contemporary control).

Supplementary Table 3. Number of individuals with a history of each mental health outcome in the past two years before the start of follow up

Reviewer #3:

Remarks to the Author:

The authors present a retrospective cohort study using UK Biobank data to compare the incidence of psychiatric morbidity and prescriptions for psychotropic medications among individuals infected with COVID-19. This represents a valuable contribution to the literature. A few comments which may assist the authors in strengthening their manuscript.

Thank you.

-In the methods, I would point out that the cluster of ICD codes that the authors' identify as anxiety disorders also includes trauma-related disorders and somatoform disorders as these are distinguished from anxiety disorders in the DSM and this may confuse readers. The authors do address this later in the manuscript, but might consider a brief parenthetical statement about this in the methods.

Response: We thank the reviewer for these suggestions.

Indeed, the full category of mental disorders to which ICD-10 codes F40-F48 correspond is "neurotic, stress-related and somatoform disorders," and most cases are diagnosed as F41 (other anxiety disorders, including generalized/unspecified anxiety disorder). Given the extremely broad readership of the Nature Portfolio, including those without specific psychiatric expertise, we used a relatively general term "anxiety disorders" and showed the results for each individual category, such as PTSD and panic disorders, to better inform readers. The same term "anxiety disorders (F40-F48)", which includes the diagnoses of trauma-related disorders and somatoform disorders, has also been used in previous large US-based EHR studies.^{1,2,7} We now add a brief statement in the Methods to explain this, as suggested by the reviewer: ...anxiety disorders (F40-F48, including trauma-related and somatoform disorders)...

-Prescription data was included for antidepressants, antipsychotics, benzodiazepines and opioids. I'm curious about the inclusion of opioids which are not typically used to treat any primary psychiatric disorder. The authors might address this point. I am also confused about the exclusion of lithium which is used almost exclusively to treat psychiatric illness (whereas many antidepressants are be prescribed for a range of non-psychiatric indications) and thus would represent a good candidate for inclusion in the analysis. Furthermore, my assumption is that other mood stabilizers

(e.g., anti-epileptics like valproic acid) were not included because of their use for non-psychiatric indications, but this should be addressed. In particular, their omission is confusing in light of the inclusion of opioids.

Response: We thank the reviewer for these suggestions.

We agree that opioids are primarily for pain relief and not typically for psychiatric disorders. However, common mental health disorders are associated with initiation and use of prescribed opioids in the general population.¹⁴ In addition, mental health conditions are often comorbid with chronic pain and polypharmacy (combinations of psychotropic and opioid medications) was reported to be increased in patients with neuro-psychiatric disorders.¹⁵ Increased prescription of opioids may indirectly reflect the increased overall burden of mental disorders after the SARS-CoV-2 infection. We note that opioids are also included as prescriptions for mental disorders in previous EHR-based study of psychiatric sequelae and an increased risk of opioids prescriptions were observed (HR, 1.76).³

We now include “mood stabilizers” as one class of psychotropic drugs. The mood stabilizers included lithium and anticonvulsants (e.g., carbamazepine, lamotrigine, valproate, valproic acid etc.), as suggested by reviewer. Some antipsychotics that can be used as mood stabilizers such as olanzapine and quetiapine were still classified in the class of antipsychotics as they are generally used to treat psychosis. Our updated analyses consistently suggested that the risk of prescriptions for mood stabilizers (HR: 1.41, 95%CI 1.18-1.67) and overall psychotropic medications (HR: 1.51, 95%CI 1.38-1.65) are increased after the SARS-CoV-2 infection compared with the contemporary control group without evidence of infection.

-If I understand correctly, the authors are denoting that psychiatric diagnoses they identify are new by the lack of corresponding prescriptions or diagnostic codes 1 year prior to the start of the study period. I would note in the discussion that the natural history of certain disorders might suggest that at least some of the cohort is having recurrent rather than de novo symptoms despite having not been treated/diagnosed in the prior year (e.g., MDD episodes can certainly occur with greater than a year in between).

Response: We thank the reviewer for these suggestions.

We fully agree that excluding only those with a diagnosis/prescription in the past year may not be sufficient to completely exclude those with previously diagnosed mental disorders, especially given the relapsing/remitting nature of mental disorders. In the original manuscript, we used two types of analyses to address this issue. First, we extended the washout period to 2 years for incident mental health outcomes in the sensitivity analyses and the results were consistent with the main analyses.

Second, because mental disorders tend to recur or relapse, we additionally estimated the risk of first or recurrent (prevalent) mental health outcomes (by including those with a diagnosis or record of related outcomes before the start of follow-up). The risk of first or recurrent outcomes was also increased.

In the revised version, regarding the main analyses for incident outcomes, we now extend the washout period from 1- year to 2-years. Further, we add an additional sensitivity analysis that strictly excluded those with a history of diagnoses/prescriptions in the past Five years before the start of follow-up. The results for any mental health outcome (HR: 1.30 [1.18-1.44]) were consistent with the main analyses using a 2-year washout (HR: 1.54 [1.42-1.67]). Most previous EHR studies used washout of 2 years to define incident mental health disorders or reported incident or prevalent outcomes separately.^{1-3,7}

-I am not qualified to comment on the nuances of the quantitative methodology but would recommend a biostatistician review and comment on these given the heavily quantitative nature of this manuscript.

-I am struggling to understand the result shared on line 322. I am inferring that the authors are comparing the COVID- 19 infection group to the non-infection cohort subgroup with negative COVID-19 tests. I am confused by the language “were not significant or decreased.” Please clarify.

Response: We thank the reviewer for these suggestions. We have rephrased the sentence as:

“Compared with test-negative control, the risks of first psychiatric diagnoses and any mental health related outcomes were significantly lower in the infection group. There was no significant difference for the risk of psychotropic prescriptions between groups”.

-The authors’ discussion paragraph about the finding that any respiratory infection was more strongly associated with new psychiatric diagnoses than was COVID-19 could be clearer. First of all, the authors posit that the individuals in the non-COVID respiratory infection cohort may have been sicker than the COVID-19 cohort. However, it was not clear to me why that would be the case; were the members of the non-COVID infection cohort all hospitalized? Or could they, too, have been diagnosed in the community setting? Furthermore, this discussion is combined in the same paragraph with a discussion about vaccination status and this makes it even more difficult to follow. I would encourage a reconfiguration of that element of the discussion to clarify the finding and articulate a possible explanatory hypothesis.

Response: We thank the reviewer for these suggestions.

In the original manuscript, the non-COVID respiratory tract infection groups included those

diagnosed in the hospital setting, and the COVID-19 infection group included those with infections tested in hospital and community settings. In the updated analyses, we compare the inpatient COVID-19 infection with inpatient non-COVID respiratory tract infection to control the potential biases from the severity of diseases.

Reviewer 1 suggested that it is more robust to compare the risk between those tested positive for SARS-CoV-2 infection in the hospital setting and those diagnosed with RTIs in the hospital setting, and to limit the non-COVID-19 RTIs to influenza only. However, only a small fraction of participants tested positive in the hospital setting (~2500), such analyses will lack power. In addition, the number of participants diagnosed with influenza in the hospital setting in 2017/03 and 2020/03 is also small. Given this, we now excluded the analyses of other respiratory tract infection in the revised manuscript.

We have rephrased and improved the part related to the vaccination and compared our results of one-year follow-up with previous EHR studies of vaccination in short term (≤ 6 months).

Minor Points:

-Line 90 should read: "All participants provided written informed consent at recruitment." You might also add "as described elsewhere" and provide a citation with further details about recruitment for the UK Biobank study.

Response: We have now cited publication of the UKB cohort profile.

-Line 90-91: Consider citing STROBE guidelines

Response: We have now cited publication of the STROBE guidelines.

-Line 96: Participants should be participate

Response: Done.

-Throughout the document the authors refer to the infected cohort as infection group. I believe it would read better with a "the" before infection group. I would also encourage the authors to consider more descriptive language such as "the COVID-19 infected cohort"

Response: SARS-CoV-2 is the virus that causes the infection, and the resulting disease is COVID-19. We now use the term "the infection group" throughout the manuscript. The text has been edited to ensure clarity: "the SARS-CoV-2 infection group (hereinafter referred to as the infection group)".

-Line 377: I think that the authors might mean specifically rather than “specially”

Response: “specially” has now changed to “specifically”.

-Line 378: WarrantS further research

Response: Done.

I'm looking forward to seeing this manuscript in print and believe that my comments can all be addressed through discussion and will not require any changes to the experimental design or analysis. Thank you for this important contribution.

With best regards,
Dan Shalev

Reference

- 1 Taquet, M., Luciano, S., Geddes, J. R. & Harrison, P. J. Bidirectional associations between COVID-19 and psychiatric disorder: retrospective cohort studies of 62 354 COVID-19 cases in the USA. *Lancet Psychiatry* **8**, 130-140, doi:10.1016/s2215-0366(20)30462-4 (2021).
- 2 Taquet, M., Geddes, J. R., Husain, M., Luciano, S. & Harrison, P. J. 6-month neurological and psychiatric outcomes in 236 379 survivors of COVID-19: a retrospective cohort study using electronic health records. *Lancet Psychiatry* **8**, 416-427, doi:10.1016/s2215-0366(21)00084-5 (2021).
- 3 Xie, Y., Xu, E. & Al-Aly, Z. Risks of mental health outcomes in people with covid-19: cohort study. *Bmj* **376**, e068993, doi:10.1136/bmj-2021-068993 (2022).
- 4 Al-Aly, Z., Bowe, B. & Xie, Y. Long COVID after breakthrough SARS-CoV-2 infection. *Nature medicine* **28**, 1461-1467 (2022).
- 5 Taquet, M., Dercon, Q. & Harrison, P. J. Six-month sequelae of post-vaccination SARS-CoV-2 infection: a retrospective cohort study of 10,024 breakthrough infections. *Brain, behavior, and immunity* **103**, 154-162 (2022).
- 6 Ranger, T. A. *et al.* Preexisting Neuropsychiatric Conditions and Associated Risk of Severe COVID-19 Infection and Other Acute Respiratory Infections. *JAMA Psychiatry*, doi:10.1001/jamapsychiatry.2022.3614 (2022).
- 7 Taquet, M. *et al.* Neurological and psychiatric risk trajectories after SARS-CoV-2 infection: an analysis of 2- year retrospective cohort studies including 1 284 437 patients. *Lancet*

- Psychiatry*, doi:10.1016/s2215-0366(22)00260-7 (2022).
- 8 Abel, K. M. *et al.* Association of SARS-CoV-2 Infection With Psychological Distress, Psychotropic Prescribing, Fatigue, and Sleep Problems Among UK Primary Care Patients. *JAMA Netw Open* **4**, e2134803, doi:10.1001/jamanetworkopen.2021.34803 (2021).
- 9 Antonelli, M. *et al.* Risk factors and disease profile of post-vaccination SARS-CoV-2 infection in UK users of the COVID Symptom Study app: a prospective, community-based, nested, case-control study. *The Lancet Infectious Diseases* **22**, 43-55 (2022).
- 10 Mizrahi, B. *et al.* Long covid outcomes at one year after mild SARS-CoV-2 infection: nationwide cohort study. *bmj* **380** (2023).
- 11 Agniel, D., Kohane, I. S. & Weber, G. M. Biases in electronic health record data due to processes within the healthcare system: retrospective observational study. *Bmj* **361** (2018).
- 12 Sauer, C. M. *et al.* Leveraging electronic health records for data science: Common pitfalls and how to avoid them. *The Lancet Digital Health* (2022).
- 13 Batty, G. D., Gale, C. R., Kivimäki, M., Deary, I. J. & Bell, S. Comparison of risk factor associations in UK Biobank against representative, general population based studies with conventional response rates: prospective cohort study and individual participant meta-analysis. *bmj* **368** (2020).
- 14 Sullivan, M. D., Edlund, M. J., Zhang, L., Unützer, J. & Wells, K. B. Association between mental health disorders, problem drug use, and regular prescription opioid use. *Archives of internal medicine* **166**, 2087- 2093 (2006).
- 15 Maust, D. T. *et al.* Prevalence of central nervous system-active polypharmacy among older adults with dementia in the US. *Jama* **325**, 952-961 (2021).

Decision Letter, second revision:

1st August 2023

Dear Mr Xie,

Thank you once again for your manuscript, entitled "Long-term mental health outcomes after SARS-CoV-2 infection: prospective cohort study," and for your patience during the peer review process.

Your manuscript has now been evaluated by 3 reviewers (Reviewers 2 and 3 from the previous round, and an additional reviewer - Reviewer 4 - whose expertise overlaps with that of Reviewer 1 from the previous round), and their comments are included at the end of this letter. The reviewers find your work to be of interest, and Reviewers 2 and 3 are happy with the revised manuscript, but Reviewer 4 raises some important outstanding concerns. We are very interested in the possibility of publishing your study in *Nature Human Behaviour*, but would like to consider your response to these concerns in the form of a revised manuscript before we make a decision on publication.

In your revised manuscript, we expect to see the new and adjusted analyses suggested by Reviewer 4, including: separate analyses in hospitalized vs non hospitalized patients with and without COVID-19; differentiation between symptoms concomitant or delayed from COVID-19 onset; adjustments to your analyses to rule out the roles of calendar period and any association between vaccination and psychiatric disturbance; a model splitting psychiatric diseases into different conditions with different pathophysiology bases. We also expect far greater clarity in the reasons for your analytical choices, including your choice of control. Please also add a discussion of potential mechanisms to your discussion as our reviewer requests. Reviewer 3 notes the position of the Methods section in the manuscript; however we ask that you keep the current structure as it fits with our journal formatting requirements.

Please note that we expect to receive this revised manuscript in two months, and if resubmission of the paper is substantially delayed, we may not be able to consider your work further for publication in Nature Human Behaviour.

In sum, we invite you to revise your manuscript taking into account all reviewer and editor comments. We are committed to providing a fair and constructive peer-review process. Do not hesitate to contact us if there are specific requests from the reviewers that you believe are technically impossible or unlikely to yield a meaningful outcome.

We hope to receive your revised manuscript within two months (by October 1st). I would be grateful if you could contact us as soon as possible if you foresee difficulties with meeting this target resubmission date.

- Include a "Response to the editors and reviewers" document detailing, point-by-point, how you addressed each editor and referee comment. If no action was taken to address a point, you must provide a compelling argument. When formatting this document, please respond to each reviewer comment individually, including the full text of the reviewer comment verbatim followed by your response to the individual point. This response will be used by the editors to evaluate your revision and sent back to the reviewers along with the revised manuscript.
- Highlight all changes made to your manuscript or provide us with a version that tracks changes.

[REDACTED]

We look forward to seeing the revised manuscript and thank you for the opportunity to review your work. Please do not hesitate to contact me if you have any questions or would like to discuss these

revisions further.

Sincerely,

Charlotte Payne

Charlotte Payne, PhD
Senior Editor
Nature Human Behaviour

REVIEWER COMMENTS:

Reviewer #2:

Remarks to the Author:

The authors have put a significant effort into improving the manuscript and have addressed and commented the points I raised in my previous review sufficiently. As such I do recommend this article for publication in Nature Human Behaviour.

Reviewer #3:

Remarks to the Author:

The authors have addressed the concerns raised in my initial review admirably and I believe the manuscript is now stronger and fit for publication. I appreciate their inclusion of mood stabilizers into their analysis, which I do think strengthens the argument significantly here. I also feel they addressed my comment about the natural history of episodic mental health disorders by increasing their washout period. I give kudos to the authors for their successful revision. I think the paper is fit for publication with one exception: it's not clear to me whether this is intentional, but the methods section seems to have gotten shifted to the end of the manuscript. As it stands right now, it reads introductionresultsdiscussionmethods in the file I am accessing. Unless this is a stylistic preference on the part of the journal, the authors should considering moving the methods between the introduction and results. I look forward to seeing this in print.

Dan Shalev

Reviewer #4:

Remarks to the Author:

The study is an interesting epidemiological work evaluating the risk of psychiatric conditions and treatment in patients with COVID-19 based on UK band- the dataset is of course of high value and the results of general interest – however, the analyses performed still need to be implemented in order to better highlight the relevance of a) premorbid status and vulnerability b) covid-19 severity c) psychosocial conditions and isolation d) vaccination on mental health condition- Moreover, the discussion should include a more general discussion about biological, psychological or general mechanisms explaining this association (which are might suitable of intervention..)

Specific comments:

One main limitation is the lack of information/discussion about possible confounders in mental health disorder incidence – isolation, hospitalization and severity of covid-9 but also premorbid

vulnerability are key factors related to possible outcomes and this should be underlined and take into consideration. Specifically, it is not clear how the authors are adjusting for the different comorbidities (or whether they are adjusting for total Charlson score , glomerular filtrate, hypertension). Moreover, It is not clear which was the a-priori distribution of psychiatric disease in cases and controls- it is important to exclude a priori bias due to possible association between psychiatric patients and infection.

The association with vaccination is highly questionable (even we also are firmly convinced of the hypothesis raised by the authors)- The vaccination status can be associated with the calendar period (diminishing the risk of severe psychiatric health comorbidity) and vaccination status itself could be associated in general with a lower risk of psychiatric disturbance (as subjects with psychiatric conditions might have less access/interest to vaccination). These issues should be definitively considered in the analyses if this is the focus of the work.

The adjustment for hospitalization is not adequate, as hospitalization itself is a main driver of possible outcomes including somatic and mental health outcomes- A separate analysis in hospitalized vs non hospitalized patients with and without COVID-19 is mandatory.

It is not clear which kind of infection(s) was considered as control- why do the authors did not use a group of patients with patients with flu/pulmonary disease (hospitalized vs non hospitalized)?

It is not clear the time of onset between psychiatric conditions/symptoms and COVID-19 disease. A differentiation between symptoms concomitant or delayed from onset of COVID-19 is essential to distinguish long-COVID from post covid (which have, again, different pathophysiology underpinnings-

As a very general comment, the paper is lacking of an explanation of such disorder and a possible model of analyses splitting different conditions based on pathophysiology mechanisms should be considered. Different vulnerability (premorbid conditions, psychological triggers, isolation, economic status?) and disease-specific mechanisms (severity of disease including ICU or hospital admission, vaccination status, pulmonary vs extrapulmonary manifestations, steroid treatments) might have different impact on incidence of An explained model splitting psychiatric diseases into different conditions with different pathophysiology bases is needed to improve the general quality of the manuscript.

The references should be updated and include the most important papers in the field- I suggest to evaluate /consider also the work focused more in general on mental health and neurological disorder - such as Long-term neurologic outcomes of COVID-19
<https://doi.org/10.1038/s41591-022-02001-z>

Author Rebuttal, second revision:

Response to reviewers' comments:

Reviewer #2:

Remarks to the Author:

The authors have put a significant effort into improving the manuscript and have addressed and commented the points I raised in my previous review sufficiently. As such I do recommend this article for publication in Nature Human Behaviour.

We thank the reviewer for the thorough assessment of our previous and updated paper and recognition of the importance of this study.

Reviewer #3:

Remarks to the Author:

The authors have addressed the concerns raised in my initial review admirably and I believe the manuscript is now stronger and fit for publication. I appreciate their inclusion of mood stabilizers into their analysis, which I do think strengthens the argument significantly here. I also feel they addressed my comment about the natural history of episodic mental health disorders by increasing their washout period. I give kudos to the authors for their successful revision. I think the paper is fit for publication with one exception: it's not clear to me whether this is intentional, but the methods section seems to have gotten shifted to the end of the manuscript. As it stands right now, it reads introductionresultsdiscussionmethods in the file I am accessing. Unless this is a stylistic preference on the part of the journal, the authors should considering moving the methods between the introduction and results. I look forward to seeing this in print.

Dan Shalev

We thank the reviewer for the thorough assessment of our previous and updated paper and recognition of the importance of this study. The Methods is written to comply with Nature style.

Reviewer #4:

Remarks to the Author:

The study is an interesting epidemiological work evaluating the risk of psychiatric conditions and treatment in patients with COVID-19 based on UK band- the dataset is of course of high value and the results of general interest – however, the analyses performed still need to be implemented in order to better highlight the relevance of a) premorbid status and vulnerability b) covid-19 severity c) psychosocial conditions and isolation d) vaccination on mental health condition- Moreover, the discussion should include a more general discussion about biological, psychological or general mechanisms explaining this association (which are might suitable of intervention..)

Response: We thank the reviewer for these

suggestions. Specific comments:

One main limitation is the lack of information/discussion about possible confounders in mental health disorder incidence – isolation, hospitalization and severity of covid-9 but also premorbid vulnerability are key factors related to possible outcomes and this should be underlined and take

into consideration. Specifically, it is not clear how the authors are adjusting for the different comorbidities (or whether they are adjusting for total Charlson score, glomerular filtrate, hypertension). Moreover, It is not clear which was the a-priori distribution of psychiatric disease in cases and controls- it is important to exclude a priori bias due to possible association between psychiatric patients and infection.

Response:

We thank the reviewer for these suggestions. We agree with the reviewer on the importance of adjusting for potential confounders and accounting for reverse causality, which we have addressed in the manuscript as follows:

Confounders

Regarding confounding factors, we used a series of predefined and data-driven covariates including a comprehensive set of established and suspected risk factors for COVID-19 and mental health conditions, to adjust for differences in baseline characteristics between the comparison groups. Confounders considered in the inverse probability-weighted cause-specific Cox models specifically included age, sex, ethnicity, index of multiple deprivation, smoking status, physical activity, body mass index, systolic and diastolic **blood pressure**, **estimated glomerular filtration rate**, hospital admissions, and **comorbidity** (cancer, cerebrovascular disease, chronic kidney disease, chronic obstructive pulmonary disease, congestive heart failure, dementia, diabetes, HIV/AIDS, hemiplegia, myocardial infarction, liver disease, renal disease, peripheral vascular disease, and peptic ulcer disease), as detailed in the Methods. To further reduce the risk of residual confounding and optimize adjustment of potential confounders, we included a list of data-driven clinical episodes diagnosed during patient hospitalization within one year before index date. We classified 8,651 source ICD-10 diagnosis codes into 453 disease phenotype groups (DPGs) using a validated mapping algorithm to account for premorbid vulnerability. After weighting, these potential confounders were well balanced between two comparison groups (ASMD <0.1) and index dates were fully aligned (Table 1 and Supplementary Figure 2).

For the three factors raised by the reviewer (isolation, hospitalization and severity of covid-9), we have discussed these key issues in the manuscript. First, regarding isolation and quarantine measures, nationwide lockdowns and self-isolation of the general population were quite common in the UK during the period of our study. The first two national lockdowns came into effect in England between January 2020 and January 2021, and England entered a third national lockdown on 6 January 2021. The comparison groups during the concurrent period, COVID-19 group vs. contemporary control group without infection, were assumed to have experienced similar contextual factors related to COVID-19, such as quarantine measures and other public health interventions. It should be recognized that actual isolation behaviour may vary between individuals (compliance with the nationwide lockdown), and the contemporary control setting was the optimal method for a large population-based study to minimize the effect of pandemic-related factors such as isolation and

other environmental stressors. In addition, we performed subgroup analyses by calendar period according to the timeline of coronavirus lockdowns and measures imposed by the UK government. The association between COVID-19 and subsequent mental health outcomes was consistently observed during the period when quarantine and isolation measures were relatively relaxed, suggesting that the observed association could not be fully explained by isolation measures (Figure 5). Second, to account for hospitalization and severity of covid-9, we further categorized the SARS-CoV-2 infection group as infection tested in hospital setting and community settings, and compared these infection subgroups with control groups separately. We found the risks of psychiatric sequelae were evident even in those who tested positive in the community setting, likely consisting of infected people with mild or asymptomatic symptoms of COVID-19, and were highest in those who tested positive in the hospital and may have more severe COVID-19.

To further reduce the concerns about bias and confounding, we use the settings of positive outcome control (to test whether our study reproduces established knowledge about long COVID (e.g., fatigue after COVID-19, one of the common long-COVID symptoms defined by WHO) and negative outcome control (e.g. risk of skin neoplasms after COVID-19 where no associations were expected according to previous knowledge). The successful use of negative outcome controls can reduce concerns about the presence of potential biases related to study design, analytical methods, residual confounding, and other latent sources of bias. This approach has been used in previous epidemiological studies.¹ Results showed that, compared with the contemporary control group without evidence of infection, the infection group had an increased risk of fatigue (HR, 2.76, 95% CI 2.38– 3.20) and dyspnea (HR, 2.98, 95% CI 2.59–3.44). No significant associations between COVID-19 and any of the negative outcome controls were observed (neoplasms of skin (HR, 0.86, 95% CI 0.69–1.06); Skin cyst (HR, 1.41, 95% CI 0.91–2.20)).

Overall, we have made good faith efforts to control potential biases and provide robust risk estimates, including the use of both predefined and data-driven covariates to adjust for the difference in baseline characteristics between comparison groups, the use of a large-scale, prospective, population-based cohort with long-term follow up, and the use of robust and accurate definitions of exposure and outcomes. Additionally, our results were robust in multiple sensitivity analyses and survived the rigorous application of negative controls. As with all observational studies, it is impossible to eliminate all confounding, which we acknowledge in the limitations section, and we do not recommend making causal inferences from ours or any other observational studies of long COVID. Nevertheless, given the lack of robust Mendelian randomization studies due to inadequate instrumental variables associated with SARS-CoV-2 infection, observational studies based on large population cohorts provide an informative reference.

Reverse causality

To account for potential reverse causality (from mental disorders to infection), we excluded

participants with diagnosis/prescription in the past two years at baseline. Most previous EHR studies used washout of 2 years to define incident mental health disorders or reported incident or prevalent outcomes separately.¹⁻⁴ Further, we add an additional sensitivity analysis that strictly excluded those with a history of diagnoses/prescriptions in the past Five years before the start of follow-up. The results for any mental health outcome (HR: 1.30 [1.18-1.44]) were consistent with the main analyses using a 2-year washout (HR: 1.54 [1.42-1.67]).

The association with vaccination is highly questionable (even we also are firmly convinced of the hypothesis raised by the authors)- The vaccination status can be associated with the calendar period (diminishing the risk of severe psychiatric health comorbidity) and vaccination status itself could be associated in general with a lower risk of psychiatric disturbance (as subjects with psychiatric conditions might have less access/interest to vaccination). These issues should be definitively considered in the analyses if this is the focus of the work.

Response:

We thank the reviewer for these suggestions. We compared the risk of the breakthrough infection (BTI, fully 2-dose vaccinated at index date) and non-BTI with non-infection cohort separately, intentionally to avoid the issues of calendar period raised by the reviewer, and the start of follow-up (index date) was aligned for both comparisons as described in our methods and results. A comparable approach has been used by Xie & Ziyad Al-Aly et al previously.⁵ We also excluded those with psychiatric diagnosis/prescription in the past two years to limit the potential influence of mental disorders on vaccination.

Of note, considering the potential impact of any health behaviors related to SARS-CoV-2 vaccinations, we restricted our comparisons to the individuals with same vaccination status. For example, when studying the risk relating to BTI, we defined control group as those who were not infected but had been fully vaccinated before the assigned index date. Our results indicated that full vaccination before infection may significantly reduce the overall psychiatric sequelae of COVID-19 as proxied by the composite psychiatric diagnosis or psychotropic prescription during the 1 year of follow-up, which is consistent with previous findings from EHR studies with shorter follow-up period. However, we dissected this overall estimate further and, for the first time, reported that the persistent psychiatric risk after BTI is likely driven by psychotropic prescriptions in the primary care (HR: 1.42 [1.00-2.02]) but not severe one diagnosed in the secondary care (HR: 0.91 [0.78-1.07]). We believe this in-depth finding has significant implications for public health interventions post-pandemic.

The adjustment for hospitalization is not adequate, as hospitalization itself is a main driver of possible outcomes including somatic and mental health outcomes- A separate analysis in hospitalized vs non-hospitalized patients with and without COVID-19 is mandatory.

Response:

We thank the reviewer for these suggestions. To ensure clarity, we'd like to highlight that our original manuscript have not only adjusted for hospitalization but also included a secondary analysis, where we specifically estimate the risk for both hospitalized and non-hospitalized COVID-19 patients (Fig.6 b) separately. Here, as required by the reviewer, we also conducted a direct comparison between hospitalized and non-hospitalized COVID-19 patients and identified a higher risk of first psychiatric diagnoses (HR, 1.40 [1.04-1.87]), psychotropic prescriptions (HR, 1.74 [1.24-2.43]), and any mental health related outcomes (HR, 1.65 [1.22-2.25]), which were consistent with the main analyses stratified by test setting.

It is not clear which kind of infection(s) was considered as control- why do the authors did not use a group of patients with patients with flu/pulmonary disease (hospitalized vs non hospitalized)?

Response:

We thank the reviewer for these suggestions. As detailed in the methods, the control groups included individuals with no evidence of SARS-CoV-2 infection (those not in the infected group who had negative testing results or never tested), with index dates were fully aligned with the infection cohort.

Given that both changes introduced by the pandemic and the direct biological effects of COVID-19 can impact mental health outcomes, we specially designed two controls to detect any potential spurious bias:

1. a contemporary control including people enrolled at the same time without evidence of infection, but were contemporaneously exposed to the broader pandemic-related changes, and
2. a historical control including people predated the pandemic enrolled at the same distribution of calendar months.

The main results were consistent vs both controls.

As for the use of flu patients as the control, in the UK Biobank, the non-COVID respiratory tract infection groups included those diagnosed in the hospital setting, and the COVID-19 infection group included those with infections tested in hospital and community settings. We agree with the reviewer that a head-to-head comparison between hospital-based COVID-19 and hospital-based other respiratory infections might be useful. However, only a small fraction of participants tested positive in the hospital setting (~10%), limiting the statistical power of such analyses. In addition, only a small number of influenza cases can be identified in our cohort as a contemporary control. Given these factors, we decided to not use participants with other respiratory infections as an additional control group.

It is not clear the time of onset between psychiatric conditions/symptoms and COVID-19 disease. A differentiation between symptoms concomitant or delayed from onset of COVID-19 is essential to distinguish long-COVID from post covid (which have, again, different pathophysiology underpinnings-

Response:

We thank for the reviewer for the comment. The primary focus of this study was to characterize the population risk and burden of mental health outcomes within 1 year after the initial infection. While we agree that differentiating between concomitant and delayed symptoms is also of clinical interest, it is beyond the scope of our research. Addressing this would require bespoke designs to account for attribution bias and data sources, such as patient-generated data, in order to provide a reliable answer.

As a very general comment, the paper is lacking of an explanation of such disorder and a possible model of analyses splitting different conditions based on pathophysiology mechanisms should be considered. Different vulnerability (premorbid conditions, psychosocial triggers, isolation, economic status?) and disease-specific mechanisms (severity of disease including ICU or hospital admission, vaccination status, pulmonary vs extrapulmonary manifestations, steroid treatments) might have different impact on incidence of An explained model splitting psychiatric diseases into different conditions with different pathophysiology bases is needed to improve the general quality of the manuscript.

Response:

We thank the reviewer for these suggestions. We did spend quite a lot space in the Discussion to illustrate several plausible mechanisms at both biological and environmental levels underline the association between COVID-19 and subsequent mental health (please see lines 281-300):

“COVID-19 may impact subsequent mental health directly and indirectly through several plausible mechanisms at both biological and environmental levels. Stress-evoking and disruptive societal changes during the pandemic may indirectly have detrimental impact on mental health in the general population.¹⁸ Specifically, the COVID-19 pandemic and related public health interventions such as quarantine and social distancing may have long-term adverse mental health consequences, especially on vulnerable groups including patients diagnosed with COVID-19 and those with pre-existing mental disorders.^{17,19} These disproportional impacts may partly explain the increased risk of mental health outcomes after infection compared with the contemporary control, although participants in both groups experienced similar pandemic-related socioeconomic and environmental stressors. In addition, possible changes in behaviours such as decreased physical activity, having a poor diet, and increased avoidance of health care and social contact in some individuals following recovery from acute COVID-19 may also contribute to the increased risk of long-term psychiatric sequelae.^{2,19} Overlapping biological factors between viral infection and

psychiatric disorders may also be implicated. Several possible underlying mechanisms include increased blood- brain barrier permeability and the central nervous system infiltration of SARS-CoV-2, chronic systemic immuno- inflammatory responses, dysregulation of microglia and astrocytes, and disturbances in synaptic signaling of upper layer excitatory neurons.^{20,21} Future studies are needed to explore whether post-COVID-19 psychiatric disorders result from SARS-CoV-2 infection itself, the disproportional adverse effects of pandemic-related factors, or a combination of both.”

We used standardized ICD-10 classification of mental and behavioural disorders which categorized disorders into 10 types according to major common themes or descriptive likeness (Supplement Table 1), which has been similarly used in previous EHR-based studies of long COVID. The observed effect estimates were relatively consistent across psychiatric subcategories in our study (range of HRs, 1.44-1.83) as well as in previous EHR studies (range of HRs, 1.35-1.94¹ or 1.78-2.03²). The current summarized evidence also did not report heterogeneous effect of the numerous psychosocial and neurobiological mechanisms underlying neuropsychiatric sequelae of COVID-19 on different disorders.^{6,7} Although we discussed potential mechanisms as detailed above, as non-experimental population-based epidemiological study, we are not able to assess whether these factors have different impact on different mental disorders. Whether biological and environmental factors have different role in the psychiatric sequelae of COVID-19 may be investigated in future studies with alternative design and analytic methods, such as genetic, metabolomic, and proteomic profiling of long-COVID mental health symptoms.

The references should be updated and include the most important papers in the field- I suggest to evaluate /consider also the work focused more in general on mental health and neurological disorder - such as Long-term neurologic outcomes of COVID-19
<https://doi.org/10.1038/s41591-022-02001-z->

Response:

We thank the reviewer for these suggestions. We have updated the reference list and included this paper on neurological outcomes.

Reference

- 1 Xie, Y., Xu, E. & Al-Aly, Z. Risks of mental health outcomes in people with covid-19: cohort study. *Bmj* **376**, e068993, doi:10.1136/bmj-2021-068993 (2022).
- 2 Taquet, M., Geddes, J. R., Husain, M., Luciano, S. & Harrison, P. J. 6-month neurological and psychiatric outcomes in 236 379 survivors of COVID-19: a retrospective cohort study using electronic health records. *Lancet Psychiatry* **8**, 416-427, doi:10.1016/s2215-0366(21)00084-5 (2021).

- 3 Taquet, M., Luciano, S., Geddes, J. R. & Harrison, P. J. Bidirectional associations between COVID-19 and psychiatric disorder: retrospective cohort studies of 62 354 COVID-19 cases in the USA. *Lancet Psychiatry* **8**, 130-140, doi:10.1016/s2215-0366(20)30462-4 (2021).
- 4 Taquet, M. *et al.* Neurological and psychiatric risk trajectories after SARS-CoV-2 infection: an analysis of 2- year retrospective cohort studies including 1 284 437 patients. *Lancet Psychiatry*, doi:10.1016/s2215- 0366(22)00260-7 (2022).
- 5 Al-Aly, Z., Bowe, B. & Xie, Y. Long COVID after breakthrough SARS-CoV-2 infection. *Nature medicine* **28**, 1461-1467 (2022).
- 6 Penninx, B. W., Benros, M. E., Klein, R. S. & Vinkers, C. H. How COVID-19 shaped mental health: from infection to pandemic effects. *Nature medicine* **28**, 2027-2037 (2022).
- 7 Davis, H. E., McCorkell, L., Vogel, J. M. & Topol, E. J. Long COVID: major findings, mechanisms and recommendations. *Nature Reviews Microbiology* **21**, 133-146 (2023).

Decision Letter, third revision:

14th February 2024

Dear Dr. Xie,

Thank you for your patience as we've prepared the guidelines for final submission of your Nature Human Behaviour manuscript, "Long-term mental health outcomes after SARS-CoV-2 infection: prospective cohort study" (NATHUMBEHAV-22102691C). Please carefully follow the step-by-step instructions provided in the attached file, and add a response in each row of the table to indicate the changes that you have made. Please also address the additional marked-up edits we have proposed within the reporting summary. Ensuring that each point is addressed will help to ensure that your revised manuscript can be swiftly handed over to our production team.

We would hope to receive your revised paper, with all of the requested files and forms within two-three weeks. Please get in contact with us if you anticipate delays.

Nature Human Behaviour offers a Transparent Peer Review option for new original research manuscripts submitted after December 1st, 2019. As part of this initiative, we encourage our authors to support increased transparency into the peer review process by agreeing to have the reviewer comments, author rebuttal letters, and editorial decision letters published as a Supplementary item. When you submit your final files please clearly state in your cover letter whether or not you would like to participate in this initiative. Please note that failure to state your

preference will result in delays in accepting your manuscript for publication.

In recognition of the time and expertise our reviewers provide to Nature Human Behaviour's editorial process, we would like to formally acknowledge their contribution to the external peer review of your manuscript entitled "Long-term mental health outcomes after SARS-CoV-2 infection: prospective cohort study". For those reviewers who give their assent, we will be publishing their names alongside the published article.

Cover suggestions

We welcome submissions of artwork for consideration for our cover. For more information, please see our guide for cover artwork.

ORCID

Non-corresponding authors do not have to link their ORCIDs but are encouraged to do so. Please note that it will not be possible to add/modify ORCIDs at proof. Thus, please let your co-authors know that if they wish to have their ORCID added to the paper they must follow the procedure described in the following link prior to acceptance:

Nature Human Behaviour has now transitioned to a unified Rights Collection system which will allow our Author Services team to quickly and easily collect the rights and permissions required to publish your work. Approximately 10 days after your paper is formally accepted, you will receive an email in providing you with a link to complete the grant of rights. If your paper is eligible for Open Access, our Author Services team will also be in touch regarding any additional information that may be required to arrange payment for your article.

Please note that *Nature Human Behaviour* is a Transformative Journal (TJ). Authors may publish their research with us through the traditional subscription access route or make their paper immediately open access through payment of an article-processing charge (APC). Authors will not be required to make a final decision about access to their article until it has been accepted. Find out more about Transformative Journals

[REDACTED]

Best regards,
Alex McKay
Editorial Assistant
Nature Human Behaviour

On behalf of

Charlotte Payne

Charlotte Payne, PhD
Senior Editor
Nature Human Behaviour

Reviewer #4:
None

Final Decision Letter:

Dear Mr Xie,

We are pleased to inform you that your Article "Long-term risk of psychiatric disorder and psychotropic prescription after SARS-CoV-2 infection among UK general population", has now been accepted for publication in Nature Human Behaviour.

Please note that *Nature Human Behaviour* is a Transformative Journal (TJ). Authors may publish their research with us through the traditional subscription access route or make their paper immediately open access through payment of an article-processing charge (APC). Authors will not be required to make a final decision about access to their article until it has been accepted. Find out more about Transformative Journals

With best regards,

Charlotte Payne

Charlotte Payne, PhD
Senior Editor
Nature Human Behaviour